# ImmGen report: sexual dimorphism in the immune system transcriptome

Shani Talia Gal-Oz [1], Barbara Maier[2], Hideyuki Yoshida[3,4], Kumba Seddu[3], Nitzan Elbaz[1], Charles Czysz[5], Or Zuk[6], Barbara E. Stranger [5,7], Hadas Ner-Gaon [1] & Tal Shay [1]*

Sexual dimorphism in the mammalian immune system is manifested as more frequent and severe infectious diseases in males and, on the other hand, higher rates of autoimmune disease in females, yet insights underlying those differences are still lacking. Here we characterize sex differences in the immune system by RNA and ATAC sequence profiling of untreated and interferon-induced immune cell types in male and female mice. We detect very few differentially expressed genes between male and female immune cells except in macrophages from three different tissues. Accordingly, very few genomic regions display differences in accessibility between sexes. Transcriptional sexual dimorphism in macrophages is mediated by genes of innate immune pathways, and increases after interferon stimulation. Thus, the stronger immune response of females may be due to more activated innate immune pathways prior to pathogen invasion.

[1] Department of Life Sciences, Ben-Gurion University of the Negev, Beer-Sheva, Israel. [2] The Precision Immunology Institute & Department of Oncological Sciences, Icahn School of Medicine at Mount Sinai, New York, NY, USA. [3] Department of Immunology, Harvard Medical School, Boston, MA, USA. [4] YCI Laboratory for Immunological Transcriptomics, RIKEN Center for Integrative Medical Sciences, Kanagawa, Japan. [5] Section of Genetic Medicine, Institute for Genomics and Systems Biology, University of Chicago, Chicago, IL, USA. [6] Department of Statistics, The Hebrew University of Jerusalem, Jerusalem, Israel. [7] Center for Genetic Medicine, Department of Pharmacology, Feinberg School of Medicine, Northwestern University, Chicago, IL, USA. *email: talshay@bgu.ac.il

The mammalian immune system displays widespread sexual dimorphism—the difference between males and females. In general, females are healthier than males, and they have better outcomes for illnesses caused by infectious diseases, sepsis, trauma, or injury[1–4]. At the molecular level, the immune responsiveness of females is higher than that of males, as it is manifested by higher levels of immunoglobulins IgM and IgG and stronger humoral and cell-mediated immunity[5,6]. Males experience infectious diseases more frequently and with increased severity[5], but females are more prone to the majority of autoimmune diseases[7,8]. In addition, there is a sex bias in the frequency and survival for various types of cancer[9], and males and females respond differently to viral vaccines[10] and transplantations[11]. Sexual dimorphism of the immune system is not limited to mammals but extends to birds, where the heterogametic sex is female, but still mortality rates from infectious diseases are higher in males[12].

There are several possible explanations for the disparity between male and female immunity. First, different sex hormones play a role in the transcriptional regulation of the immune system[13]. Second, different composition of sex chromosomes provides males with Y chromosome genes and females with two potentially different copies of X-linked genes[14]. Cell-specific X chromosome inactivation results in female-specific mosaicism. However, some genes of the X chromosome escape the inactivation, resulting in higher expression levels of specific genes in females[15]. It is possible that systemic or tissue-specific X inactivation or escape from inactivation, together with X- and Y-linked immune-related genes, cause immunological differences between males and females[14]. For example, the X-linked gene *Tlr7* plays a role in innate immune response and displays higher expression in females compared to males, potentially due to incomplete X-inactivation[16]. Another example of this potential effect is the X-linked gene *Ddx3x* and its Y-linked homolog *Ddx3y*. *Ddx3x* is crucial for interferon (IFN) production in response to pathogens[17] and in high levels can boost the female IFN-inducer response. Indeed, mice lacking *Ddx3x* in hematopoietic cells have higher susceptibility to *Listeria monocytogenes* and reduced numbers of lymphocytes, not compensated by *Ddx3y*[18]. Third, stimulations related to pregnancy and breastfeeding are limited to females. Pregnancy depresses maternal immunity as a means of preventing rejection of the fetus[19]. Fourth, there are behavioral differences between males and females, with regard to the frequency and timing of exposure to challenges (viral, bacterial, chemical, trauma, and others), and food intake prioritization in the family[6]. Nonetheless, hormones, motherhood, and social differences cannot be the only mechanisms, as immunity differences are evident in a variety of other situations, namely, prior to sexual maturity—for example, the sex bias in some autoimmune diseases in pediatric patients, in whom the expression of sex hormones is low[8], in women who have not experienced pregnancy, and also in mice. To date, the relative contribution (if any) of the explanations above is unknown.

One of the most striking sex differences in autoimmune diseases is of systemic lupus erythematosus (SLE), in which nine of the ten patients are female[8]. SLE is characterized by a wide profile of autoantibodies, causing inflammation and irreversible organ damage[20]. Overproduction of IFNs, a group of signaling proteins in the immune system, is highly correlated with autoimmunity[21], particularly with SLE[22]. Yet, the molecular mechanism explaining the higher rates of SLE in women is unknown. Thus we hypothesize that sex-based differential activation of IFN pathways may contribute to sexual dimorphism in SLE and other autoimmune disorders.

Sexual differential gene expression in mammals is being studied to explain phenotypic differences between the sexes[23]. Significant transcriptional differences of autosomal genes between sexes were identified not only in reproductive organs[24,25] but also in other organs[23,24,26]. There are conflicting findings regarding sexual differential expression in whole blood, ranging from many differentially expressed genes[27] to none[26]. Considering that immune cell type frequencies differ between sexes[28] and the transcriptional differences between immune cell types[29], whole blood is expected to be different between sexes. However, it is unclear whether the difference will remain when each cell type is compared separately. One study aimed to regress out the frequencies of cell types to identify cell-type-specific differences[30], but such decomposition is of limited accuracy. Specific genes have been shown to be differentially expressed between sexes in specific cell types. For example, *Pparα* mRNA expression was higher in male compared with that in female CD4$^+$ T cells in several mouse strains[31]. Nonetheless, to date, a systematic study of transcriptional sexual dimorphism of the immune system across several cell types has not been undertaken in either human or mouse.

To the best of our knowledge, cell-type-specific sex effect on transcriptome has been studied in the immune system only for bone marrow-derived macrophages (BMDM)[12,32] and microglia—the macrophages of the central nervous system (CNS). Microglia exhibit a small number of differentially expressed genes, which are mainly located on the sex chromosomes[33]. In murine BMDM from AKR and DBA/2 F2 cross, 6719 transcripts were found to be differentially expressed between sexes, but only 4% of those with a fold change >2[32]. In chicken BMDM, IFN-inducible genes expression is higher in female than in male[12], even though the heterogametic sex in chickens and all birds is female (ZW), and the IFN-α and IFN-β clusters are located on the Z chromosome, of which males have two copies (ZZ).

The Immunological Genome Project (ImmGen) aims to create a comprehensive map of the transcriptome of the immune system of the mouse and its regulation. Until now, the map focused on male mice. Here we extend the map to include female mice. We profile the transcriptomes of 11 unstimulated and 3 IFN-induced immune cell types in male and female mice to map the transcriptional sexual dimorphism of the immune system and to identify factors that contribute to the observed differences in disease prevalence between the sexes. To the best of our knowledge, this study is the first to explore overall immune transcriptional and regulatory sexual dimorphism at the baseline and after immune stimulation. Thus it provides a starting point to identify transcriptional changes underlying the phenotypical changes between the male and female immune responses.

## Results

**Transcriptional profiling**. To identify immune transcriptome sexual dimorphism, we analyzed RNA sequencing (RNA-seq) profiles from the 11 immune cell types comprising the ImmGen 11 cell set from male and female C56BL/6J mice. This 11 cell set encompasses all the major immunocyte lineages: granulocytes (GNs), dendritic cells (DCs), macrophages (MFs), B1a and B2 B cells (B), CD4$^+$ (T4) and CD8$^+$ (T8) T cells, regulatory T (Treg) cells, natural killer (NK) and natural killer T (NKT) cells, and gamma delta T (Tgd) cells. A total of 183 samples (92 females and 91 males) were profiled in four datasets differing in protocols, unstimulated or IFN stimulated, ages, and cell types (datasets A–D, see "Methods"; Fig. 1a, Supplementary Data 1).

Overall, the largest effect on the transcriptional profile was exerted by the cell type, which overpowered the sex effect. An

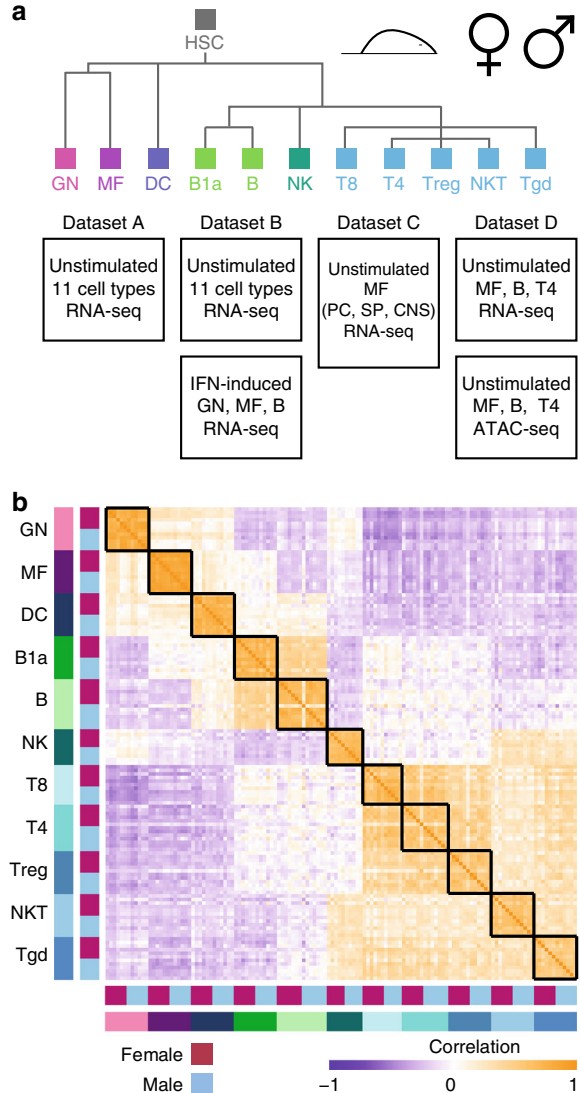

**Fig. 1** Cell-type transcriptional signature masks the sex effect. **a** Overview of the datasets used in this study. All four datasets include samples from male and female mice, from some or all of ImmGen 11 cell set, whose cell types are shown in the tree on the top. On the bottom, details of the four datasets used. PC peritoneal cavity, Sp spleen, CNS central nervous system, ATAC-seq Assay for Transposase-accessible Chromatin, GN granulocytes, DC dendritic cells, MF macrophages, T4 CD4+ T cells, T8 CD8+ T cells, Treg regulatory T cells, NK natural killer cells, NKT natural killer T cells, Tgd gamma delta T cells. **b** Pearson correlation matrix of the samples comprising datasets A and B (including the entire 11 cell set) for the 2904 genes that are expressed in both datasets, sorted by cell type and sex. Black rectangles mark samples of the same cell type. Source data for **b** are provided as a Source Data file

examination of a correlation matrix between the 11 unstimulated cell types (66 female and 66 male samples from datasets A and B, Fig. 1b) showed the highest correlations between samples of the same cell type (~0.5–1). Lineage-related samples displayed a lower—but still strong—correlation. The structure of the correlation matrix thus matched the immune lineage topology (Fig. 1a, b) and was similar to that reported in previous transcriptional studies[29]. Similar results can be seen in the principal component analysis (PCA) of all samples (Supplementary Fig. 1a). The difference between cell types constitutes the

largest source of variation in the data, covered by at least the first three principal components (explaining 28%, 17%, and 11% of the total variance), and thus masks the effect of all other factors (dataset, age, and sex). Principal component 12 is the first to separate samples by sex (paired $t$ test, pFDR $1.4 \times 10^{-6}$, paired by dataset and cell type), explaining 1.04% of the variance (Supplementary Fig. 1b). Out of the 56 genes contributing to this principal component (PCA coefficient $|>0.05|$, Supplementary Table 1), only 4 are on the sex chromosomes, suggesting that the transcriptional sexual dimorphism in the immune system is not limited to the sex chromosomes.

**Pan-immune sex signature.** To address the challenge of identifying subtle differences between male and female immune transcriptomes, a pan-immune approach was designed to identify a consistent effect, possibly of small magnitude, across all cell types. The pan-immune approach was implemented by applying a paired $t$ test to the RNA-seq datasets described above (66 female and 66 male samples from 11 unstimulated cell types of datasets A and B, Fig. 1a), in which samples were matched by cell type, age, and dataset. Datasets A and B were produced independently and are different in the ages of the samples, the protocols, and the depth of sequencing. The pairing was done by dataset and age, to account for batch effects and age-related changes, respectively. Though age-related changes could be sex dependent, the number of samples in each age group does not allow identification of such effect accurately.

Only 14 sexually differentially expressed genes (SDEGs) were identified [false discovery rate (FDR)-adjusted paired two-sided $t$ test $p$ value (pFDR) < 0.2, female–male fold change >1.5; Fig. 2a, Supplementary Table 2]. Among those 14 genes, expression was higher for 3 genes in females (including *Xist*, X-linked) and for 11 genes in males (including *Eif2s3y*, Y-linked). Four of the SDEGs contributed to the sex separation of principal component 12 (Supplementary Fig. 1b), with the female SDEGs *Oas3* and *Xist* and male SDEGs *Abcg1* and *Eifs3y*.

Since not many individual genes passed the above criterion for pan-immune SDEGs, we assessed whether pathway analysis could identify enrichment in our sorted list of the male–female difference of all genes. Indeed, several gene sets were significantly enriched in the top of the list, that is genes most upregulated in females [paired $t$ test (pairing by cell type, dataset and age) followed by pre-ranked gene set enrichment analysis (GSEA)[34], default GSEA threshold (FDR < 25%); Fig. 2b, Supplementary Table 3], all for immune-response-related pathways: IFNα and IFNγ response, complement and coagulation and Janus-activated kinase (JAK)-signal transducer and activator of transcription factor 3 (STAT3) signaling. As most of the variance in the data was found to be lineage based (Fig. 1b) and not many SDEGs were identified at the pan-immune level (Fig. 2a), we hypothesized that differences in gene expression are cell-type specific and followed different strategies to identify them.

**Cell-type-specific sex signature.** Cell-type-specific immune transcriptional sexual dimorphism is characterized by differences in gene expression between sexes that are limited to one or few types of immune cells. Paired $t$ tests (paired by dataset and age) were applied to male and female samples for each of the 11 cell types separately. Similar to the pan-immune analysis, pairing was done by dataset and age.

As expected, the genes *Xist* and *Eif2s3y* (located on X and Y chromosomes, respectively) identified in the pan-immune analysis were differentially expressed in most (9/11) cell types. The pan-immune SDEGs were not necessarily identified in each

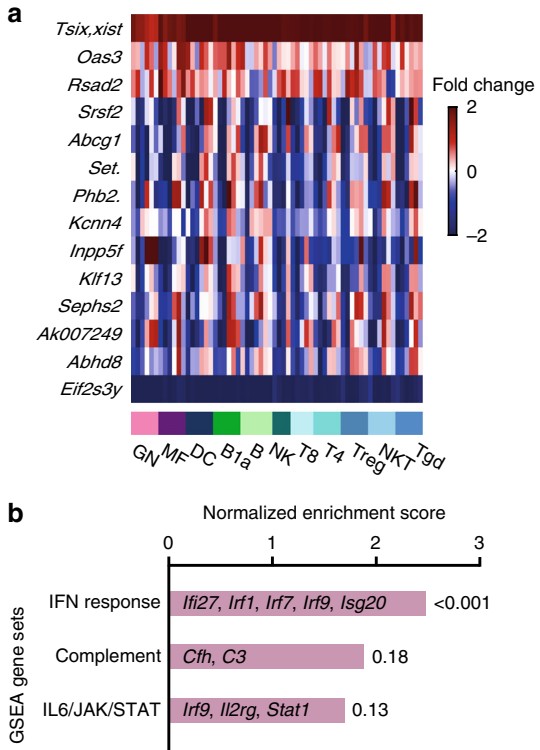

**Fig. 2** Pan-immune transcriptional sexual dimorphism. **a** Heatmap of $\log_2$ fold change (female–male expression) of pan-immune female and male sexually differentially expressed genes (SDEGs) (datasets A and B, paired $t$ test pFDR < 0.2, female–male fold change >1.5). For genes with multiple names, only the first is shown and then period. Source data included in Supplementary Table 2. **b** Female enriched pathways identified by pre-ranked gene set enrichment analysis [GSEAPreranked[34] (FDR < 25%) of the $t$ statistic, sorted by normalized enrichment scores]. FDR corrected $p$ value is displayed to the right of each bar. Source data included in Supplementary Table 3

cell type, as the much lower number of samples in each cell type reduce the power of the test (Supplementary Note 1).

Except for a single autosomal gene, *Rps17*, higher in male than in female Treg cells, autosomal cell-type-specific SDEGs were only identified in MFs (Fig. 3a). Among the 41 SDEGs identified in MFs (pFDR < 0.2, female–male fold change >1.5, permutation $p$ 0.034), the expression of 26 (63%) was higher in females. Those include genes involved in the complement system (*Fcgr2b, Fcgr3A, Lrg1, Cfb, Prtn3, Xdh, Bst1,* and *C4a/ C4b*) and also in lipoprotein metabolism and the statin pathway (*Apoe, Apoc2/Apoc4,* and *Pltp*). Expression of 15 genes (36%) was higher in males, including one Y-linked gene (*Eif3Sy*) (Fig. 3b, Supplementary Table 4, Supplementary Note 2, Supplementary Fig. 2a, b).

As we hypothesized above that the immune sexual dimorphism may partially result from sex-based differential activation or sensitivity of IFN pathways, and transcriptional dimorphism is mostly evident in macrophages, we sought to assess how IFN-related immune pathways differ between male and female unstimulated MFs. For that, we examined the expression of IFN-stimulated genes in MF list (MF-ISGs) from Mostafavi et al.[35] in unstimulated MFs (Fig. 3c). Whereas 7 of the 26 female MF-SDEGs are also MF-ISGs, none of the 15 male MF-SDEGs is an MF-ISG. Thus some of the genes involved in IFN-

related immune pathways are upregulated in females, possibly contributing to higher activation of IFN pathways in unstimulated females, which, in its turn, contributes to phenotypic sex differences.

As arbitrarily defined thresholds may cause underestimation of the similarity between the female MF transcriptome and IFNα transcriptional response, we compared the fold change (female/male) distribution of the entire macrophages' IFN signature[35] with that of all other genes. While the distribution of the female–male fold change of all genes is symmetrical, that of MF-ISGs tended to the female side (right) and differed significantly from that of the distribution of all genes (two-sided $t$ test, $p$ $7.6 \times 10^{-12}$, permutation $p < 0.001$; Fig. 3d). These results suggest that the expression of IFN-response genes in female MFs tends to be higher than in male MFs in the unstimulated state, thereby possibly strengthening the immune responsiveness in females.

After demonstrating that the genes that are expressed higher in females MFs at the unstimulated state are enriched for the IFN response, we wanted to study how the IFN-response genes in general differ between sexes at the unstimulated state. The highly complicated IFN response of the same 11 cell types as here was recently studied in detail[35]. Briefly, Mostafavi et al.[35] built a regulatory network between 92 predicted regulators and 102 ISGs, with 2691 links connecting regulators to genes based mostly on co-expression in human and mouse response to IFN. The 102 ISGs in the regulatory network were then parsed into five distinct modules—C1–C5—based on the similarity of their regulators. Thus the modules are different in terms of regulation by definition and also distinct in the cell types in which they are active, their tonic IFN response, and more. For example, STAT1/2 and IRF9 were the main predicted regulators of C3 and C4, and accordingly those modules were enriched for core ISRE motif. Module C3 contained antiviral effectors and key regulators. C1 and C2 were enriched for RNA processing, C4 for metabolic regulation, and C5 for inflammation mediators or regulators[35].

We tested the unstimulated female–male fold change of the genes in each of those five modules[35]. The genes in the antiviral module C3 display significantly positive fold change values ($t$ test pFDR $4.2 \times 10^{-11}$, Fig. 3e), meaning that the antiviral response is higher in unstimulated MFs from females compared to males, in agreement with the C3 module being responsive to tonic IFN. In the unstimulated state, module C5 also show the same trend toward being higher in females but does not reach the level of significance here ($t$ test pFDR 0.082, Fig. 3e).

**Comparison to SLE datasets.** To assess whether the identified transcriptional sexual dimorphism is relevant to the phenotypic sexual dimorphism, we focused on the most sex-biased auto-immune disease, SLE. The SLE score, developed by Mostafavi et al., measures the association between diagnosis of SLE and the expression levels for each individual gene[35]. SLE scores were calculated there[35] for two SLE studies of European[36] and East Asian[37] ancestry. These studies compared the expression profiles of peripheral blood mononuclear cell (PBMC) from SLE patients with age- and sex-matched controls. We compared the female–male fold change of the MF-ISGs in unstimulated immune macrophages to those genes' SLE score. Genes with high SLE score in both datasets (Supplementary Fig. 3a, b) display higher female/male fold change in unstimulated macrophages compared to genes with lower SLE score. This suggests that some of the SLE-contributing genes display different expression levels between sexes at the baseline healthy state.

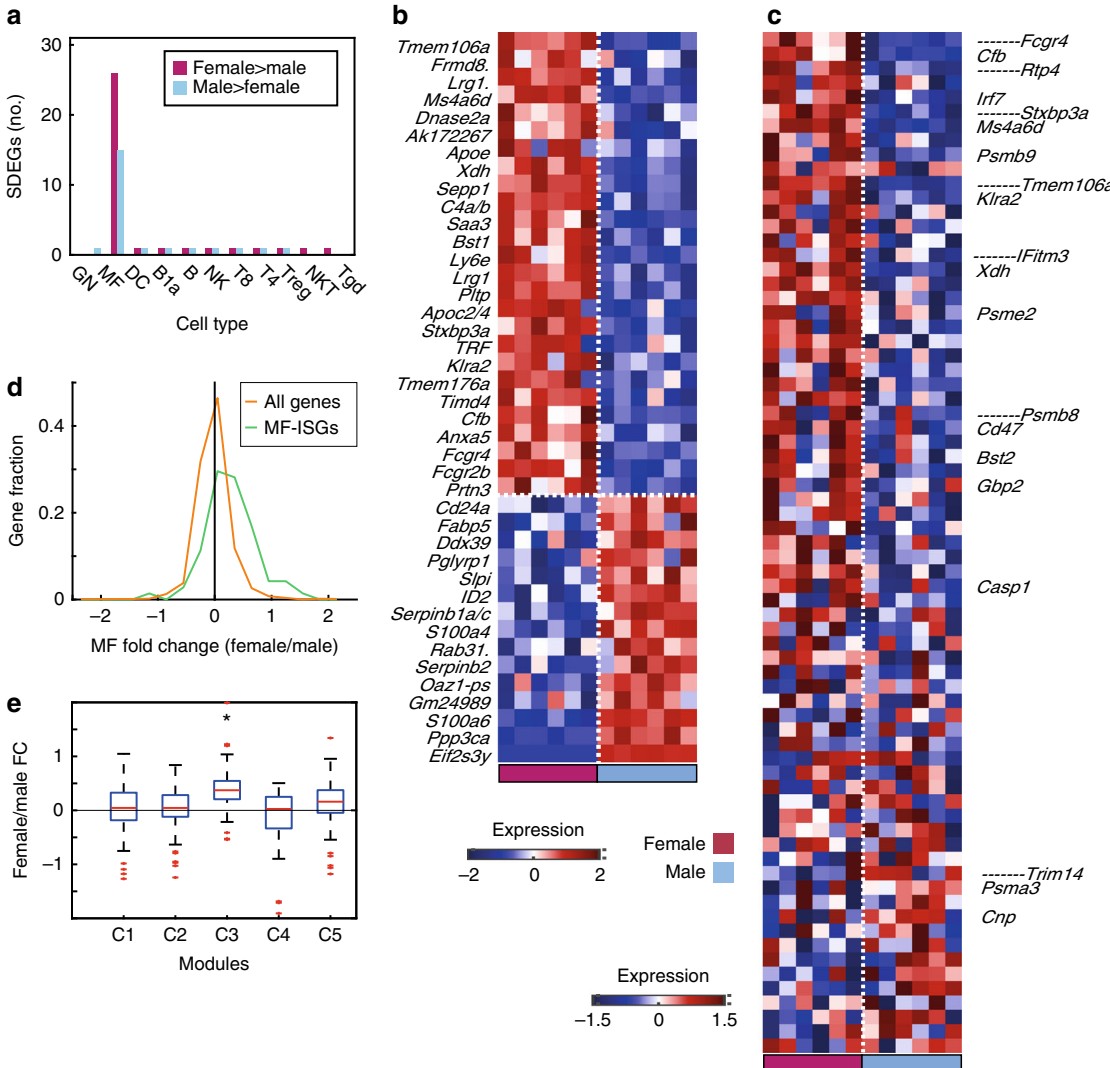

**Fig. 3** Cell-type-specific sex signature. **a** Number of male (light blue) and female (pink) sexually differentially expressed genes (SDEGs) in each cell type (datasets A and B, paired $t$ test pFDR < 0.2, female–male fold change >1.5). **b** Heatmap of relative expression levels of male and female SDEGs in MFs from datasets A and B. Genes are sorted by female–male fold change. Expression values are trimmed to range [−2, 2]. For genes with multiple names, only the first is shown and then period. **c** Heatmap of relative expression levels of 71 previously identified MF IFN-stimulated genes (MF-ISGs)[35] in datasets A and B. Gene symbols are indicated for MF-ISGs that are also SDEGs or known IFNα-response genes (according to MsigDB Hallmark gene sets). Expression values are trimmed to range [−1.5, 1.5]. In **b**, **c**, dotted horizontal white lines separate female from male upregulated genes. Dotted vertical white lines separate female and male samples. **d** Female–male fold change ($\log_2$) distribution of MF-ISGs (71 genes, green) and all genes (1568 genes, excluding MF-ISGs, orange). The $t$ test $p$ value between distributions is $7.6 \times 10^{-12}$, with permutation $p$ value <0.001. **e** Box plot presenting female–male $\log_2$ fold change (FC) distribution of the genes in modules C1–C5 defined by Mostafavi et al.[35] from unstimulated macrophages. On each box, the central mark indicates the median, and the bottom and top edges of the box indicate the 25th and 75th percentiles, respectively. The whiskers extend to the most extreme data points not considered outliers, and the outliers are plotted individually in red. Modules whose distributions are significantly different from zero (one-sample two-sided $t$ test pFDR < 0.05) are marked by asterisk. Source data for **b**–**e** are provided as a Source Data file

**Immune sexual dimorphism in the IFN response**. Although some transcriptional differences between the male and female immune systems have been already identified here in the unstimulated mice, we hypothesized that additional differences might develop or increase after immune stimulation. As unstimulated female MF-SDEGs were enriched for IFN-response genes, we initially tested this hypothesis with regard to IFN stimulation. To this end, we analyzed RNA-seq profiles of three cell types, GN, MF, and B cells, from male and female mice before and after exposure to IFNα. PCA of the data showed separation of the samples based on their cell type (PC1 and PC2, 75% of the variance; Fig. 4a). The third PC (PC3), which separated the samples

by response to IFN, accounted for 6% of the variance in the data. In GNs and MFs, the separation between males and females increased with the response to IFN along PC3 (the IFN-response axis). Overall, the genes whose expression level was upregulated in response to IFN in MF are the same between males and females (Supplementary Fig. 4a). However, the female IFN-stimulated to unstimulated fold change was generally greater compared to the male (two-sided paired $t$ test $p$ value $2.7 \times 10^{-13}$, permutation $p$ value <0.001, Fig. 4b, Supplementary Data 2). These findings suggest that IFN stimulation has greater effect on females and thus increases the transcriptional gap between male and female MFs, compared to their unstimulated state. Similar

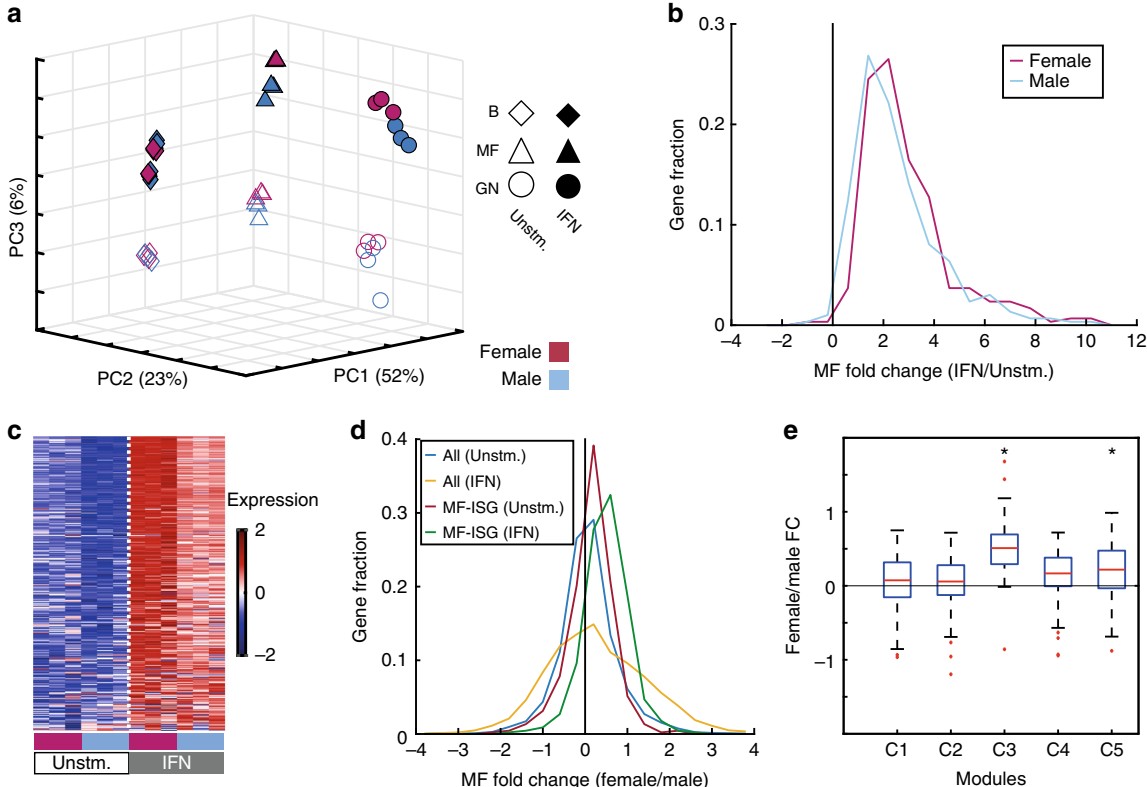

**Fig. 4** Interferon boosts the transcriptional sexually differential immune response. **a** Principal component analysis (PCA) of GN, MF, and B cells (circles, triangles, and diamonds, respectively) of females and males (pink and light blue) with and without IFN (filled or open shapes, respectively), based on 1000 autosomal genes with the highest standard deviation. Axes are the first three principal components (PCs). **b** The distribution of the $\log_2$ fold change values between IFN-stimulated MF and unstimulated MF for the IFN upregulated genes in male and/or female MF. Distribution for female samples is shown in pink and for male in light blue. The t test p value between distributions is $2.7 \times 10^{-13}$ with permutation p value <0.001. **c** Expression heatmap of 313 MF IFN-stimulated genes (MF-ISGs) expressed in unstimulated and IFN-stimulated male and female macrophage samples (dataset B). Expression values are trimmed to range [−2, 2]. Dotted vertical white line separate IFN-stimulated and unstimulated samples. **d** Female–male fold change distribution of MF-ISGs and all remaining genes from unstimulated samples (red and light blue, respectively) and MF-ISGs and all remainder genes from IFN-stimulated samples (green and yellow, respectively). **e** Box plot presenting female–male $\log_2$ fold change (FC) distribution of the genes in modules C1–C5 defined by Mostafavi et al.[35] from IFN-stimulated macrophages. On each box, the central mark indicates the median, and the bottom and top edges of the box indicate the 25th and 75th percentiles, respectively. The whiskers extend to the most extreme data points not considered outliers, and the outliers are plotted individually in red. Unstm unstimulated, IFN interferon. Modules whose distributions are significantly different from zero (one-sample two-sided t test pFDR < 0.05) are marked by asterisk. Source data for **a**–**e** are provided as a Source Data file

results were found for GNs (Supplementary Fig. 4b) but not for B cells (Supplementary Fig. 4c).

To further evaluate the behavior of the IFNα pathway in MFs, we examined the expression of the MF-ISGs[35] in unstimulated and stimulated male and female MFs (Fig. 4c). The genes were indeed induced in both sexes, but for both unstimulated and IFN-stimulated MFs, the expression level was higher in female compared to that in male. Accordingly, in IFNα-stimulated samples, the female–male fold change distribution of MF-ISGs was higher than the distribution of all other genes (two-sided t test, p value $1.5 \times 10^{-3}$, permutation p value <0.001; Fig. 4d), which means MF-ISGs as a group are upregulated in females compared to that in males. This distribution was also significantly different from the MF-ISG distribution of female–male fold change in unstimulated samples (two-sided t test p value $9.74 \times 10^{-16}$, permutation p value <0.001; Fig. 4d), meaning that the male–female difference in MF-ISGs increases upon IFN stimulation.

To characterize the sexually dimorphic genes following IFN stimulation and compare those to the baseline sexually dimorphic genes, we tested the IFN-induced female–male fold change of each of the five IFN-response modules[35] (Fig. 4e). The genes in antiviral C3 module, which were already upregulated in unstimulated females (Fig. 3e), display even bigger positive fold change values in response to IFN (one-sample two-sided t test pFDR $5.3 \times 10^{-23}$), hinting that the antiviral IFN response is more activated in females compared to that in males, and the difference increases with IFN stimulation. The genes in the inflammation module, C5, are significantly higher in females only after IFN stimulation (one-sample two-sided t test pFDR $5.6 \times 10^{-6}$), though showing this trend at the unstimulated state already, suggesting that C5 genes response to IFN increases the gap between males' and females' immune response.

**Sexual dimorphism of chromatin accessibility in immune cells.** During transcription and as a response to specific stimuli, the dense chromatin structure in particular genomic regions becomes unpacked and more accessible, allowing the entry of the factors necessary for transcription. Thus a study of open chromatin regions (OCRs) might provide information on the regulatory status of the cells. To identify OCRs, we used the ImmGen dataset

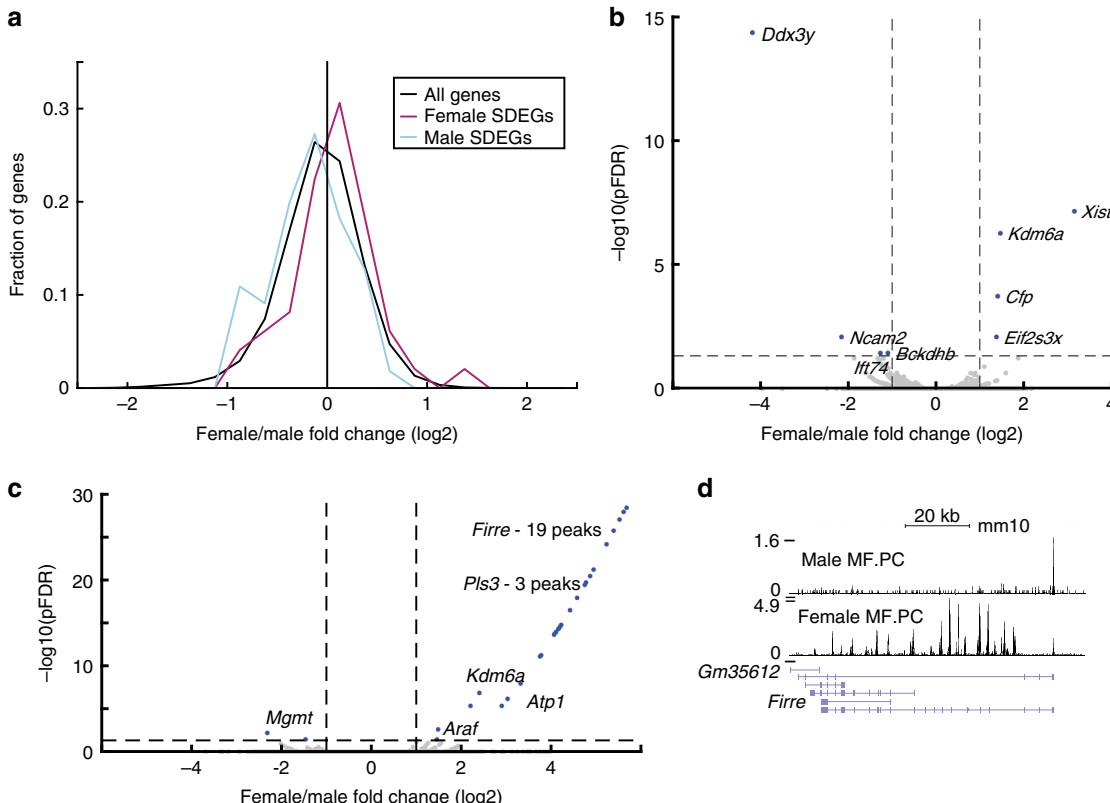

**Fig. 5** ATAC-seq data of MFs display sexual dimorphism in chromatin accessibility. **a** Female–male fold change (log$_2$ gene body OCR value) distribution of MF-specific female (pink) and male (blue) SDEGs and all other genes (termed all genes, excluding SDEGs, black). **b** Sexual differential accessibility in macrophage TSS OCRs (adjusted two-sided $t$ test pFDR < 0.05 and female–male fold change >2, shown in blue). **c** Sexual differential accessibility in macrophage distal enhancer OCRs (adjusted two-sided $t$ test pFDR < 0.05 and female–male fold change >2, shown in blue). **d** The Firre locus chromatin accessibility data on the UCSC data browser (http://rstats.immgen.org/Chromatin/chromatin.html) in MF male (top) and female (bottom). Source data for **a**–**c** are provided as a Source Data file

of Assay for Transposase-Accessible Chromatin (ATAC-seq) data[38] that profiled many immune cell types. For three cell types, namely, B, MF, and T4 cells, both male and female mice were profiled. OCRs were defined, and their association with genes, either as gene body, transcriptional start sites (TSS), or distal enhancers (DEs), was identified[38].

To estimate the accessibility of the MF-SDEGs gene body in the ATAC-seq data for MF, we compared the distributions of female–male fold change OCR binding between three sets of genes: all genes, female MF-SDEGs, and male MF-SDEGs (Supplementary Table 4, Supplementary Note 2). Gene body OCR female–male fold changes in both female and male SDEGs differed significantly from those of the remainder of the genes [two-sided $t$ test $p$ value 0.025 (female), 0.038 (male); permutation $p$ value 0.035 (female), 0.04 (male); Fig. 5a]. The matched RNA-seq of the same samples (dataset D, Fig. 1a, Supplementary Data 1) were similarly analyzed, and all MF-SDEGs were consistent with Datasets A and B (Supplementary Fig. 5), with significantly different female–male fold change distributions between female SDEGs and all genes and between male SDEGs and all genes [two-sided $t$ test $p$ value $2.7 \times 10^{-3}$ (female), $3.4 \times 10^{-10}$ (male)].

We defined a differential OCR (DO) as an OCR with higher levels of accessibility in one of the sexes compared to the other (pFDR < 0.05 and female–male fold change >2). We focused on two groups of DOs, those that fall within TSS according to RefSeq definition (TSS DOs) and those that at more distal locations,

representing DEs[38]. We identified 8, 1, and 5 TSS DOs corresponding to unique genes in MF, B, and T4 cells, respectively (Fig. 5b, Supplementary Fig. 6a, b, Supplementary Table 5). All female-specific DOs mapped to the X chromosome, including Xist (in MF and T4)[15], Kdm6a (in MF and T4), and Eif2s3x (in MFs), which are known to escape X inactivation[39]. Within MF TSS DOs, we also identified Cfp, which is the only known positive regulator of the alternative complement pathway[40]. Among the male-specific DOs, three autosomal loci were identified: Bckdhb, Ift74, and Ncam2 (in MF only).

Within the DE OCRS, 29 and 34 DOs, associated with 6 and 11 unique genes, were identified in MF and T4, respectively (Fig. 5c, Supplementary Fig. 6c, d, Supplementary Table 5), but none in B cells. Most of those DOs were female specific (93% and 88% for MF and T4, respectively). Among the female DE DOs in MF and T4 cells, two specific loci on the X chromosome aroused our curiosity. The first was the region including the long non-coding RNA Firre and an adjacent predicted gene Gm35612 (Fig. 5c, d). Most of the female MF DE DOs (19 DOs, 66%) and half of the female T4 DE DOs (17, 50%) were annotated to either Firre or Gm35612. Within the female DOs in MF and T4 cells, the second locus of interest encompasses the gene Pls3 and the adjacent long non-coding RNA 4933407K13Rik.

**Sex differences in macrophages in various tissues.** We have shown that peritoneal cavity (PC) MFs exhibit sex-biased gene expression, with female-specific upregulation of innate immune

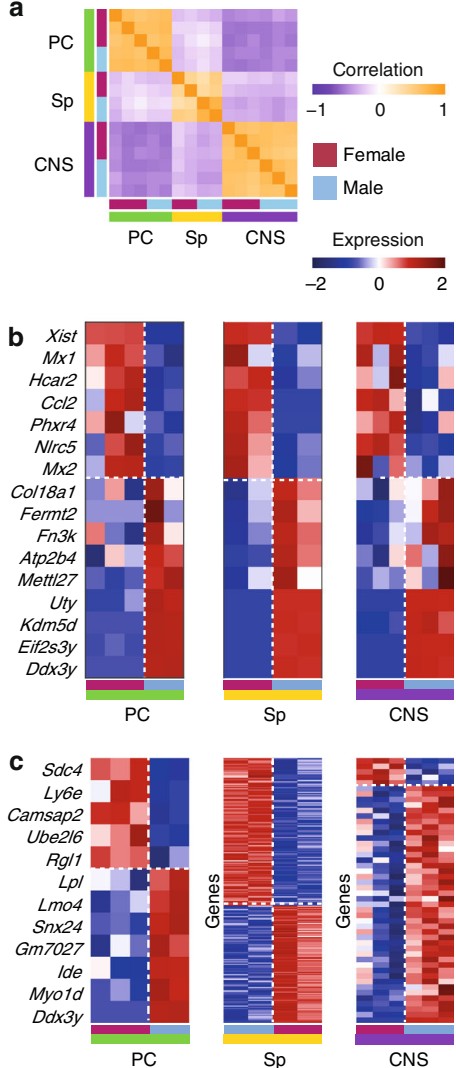

**Fig. 6** Sexual dimorphism in macrophages across tissues. **a** Pearson correlation matrix of MFs from the peritoneal cavity (PC-MF, green), spleen (Sp-MF, yellow), and microglia (labeled CNS, purple), based on 10,264 genes expressed above the noise threshold, sorted by cell type and sex. **b** Heatmap of the expression values of genes that are significantly contributing (two-way ANOVA, pFDR < 0.2, female–male fold change >$\log_2(1.5)$ in all three tissues) to sex effect across tissues (16 genes). **c** Heatmap of the expression values of genes that are significantly contributing to the sex–tissue interaction effect in specific tissue (two-way ANOVA pFDR < 0.2, male–female tissue-specific fold change >$\log_2(1.5)$): left PC-MF, middle Sp-MF, 127 female and 101 male specific genes; right microglia (labeled CNS), 5 female and 43 male specific genes. In **b**, **c**, dotted horizontal white lines separate female from male upregulated genes. Dotted vertical white lines separate female and male samples. Values were normalized per tissue. Source data for **a**–**c** are provided as a Source Data file

response genes. As phenotypic immune sexual dimorphism occurs systemically across tissues in various diseases, we extended the analysis to include not only MFs from the PC but also MFs from the spleen (Sp) and microglia, the resident MFs of the CNS (dataset C—PC-MFs: 3 males, 2 females; Sp-MFs: 2 males, 2 females; CNS: 3 males, 3 females, Fig. 1a, Supplementary Data 1). To understand the robustness of sexual dimorphism in MFs across these tissues, we first studied the expression of the MF-

SDEGs we identified above (Fig. 3b) in the three populations of tissue-resident MFs. The overall pattern of the MF-SDEGs that passed the expression filtering in each tissue is consistent in this independent set of PC-MFs of dataset C but less so in Sp-MF and microglia (Supplementary Fig. 7a–c). The strongest effect on the variance in the data is the tissue of origin, with Sp-MF similar to PC-MF and CNS, and PC-MF and CNS more distant from each other (Fig. 6a).

To pinpoint the genes responsible for the sex difference in tissue-resident MFs, we examined the effects of sex, tissue, and their interaction. The sex effect was significant for only 16 genes (two-way analysis of variance (ANOVA) sex effect pFDR < 0.2 and female–male fold change >1.5 in all three tissues; Fig. 6b, Supplementary Data 3). Expression of 7 of those 16 genes was higher in females, including the expected *Xist*. Expression of the other 9 genes was higher in males, including the Y-linked genes: *Eif3sy*, *Ddx3y*, *Uty*, and *Kdm5d*. The effect of the interaction between tissue and sex was evident in 254 genes that were differentially expressed between sexes in at least one tissue (two-way ANOVA interaction pFDR < 0.2, female–male tissue-specific fold change >1.5 and above expression threshold in at least two samples of that tissue; Supplementary Data 3). Among the interaction genes, 5 were higher in female PC cells (Fig. 6c top left), including the female SDEG, *Ly6e*. Another seven genes were higher in male PC cells (Fig. 6c bottom left; Supplementary Data 3). PC-MF-specific genes were too few to allow identification of pathway enrichment. In the Sp, 127 female-specific genes were identified (Fig. 6c top middle; Supplementary Data 3), including two that were identified also in PC-MF (*Ly6e*, *Rgl1*). These are enriched for five pathways (hypergeometric test, FDR < 0.05; Supplementary Table 6), including tumor necrosis factor-α signaling via nuclear factor (NF)-κB, P53 pathway, complement, and also the androgen response. Male Sp-MF presented 101 genes (Fig. 6c middle bottom) including 3 common with PC-MF (*Ide*, *Myo1d*, and *Ddx3y*), enriched for E2F target genes and heme metabolism (hypergeometric test, FDR < 0.05; Supplementary Table 6). In microglia, five female-specific genes were identified (Fig. 6c right top; Supplementary Data 3), including the female SDEG *Ccl2*, which was also upregulated as a response to IFN in both sexes. Another 43 microglia male-specific genes were identified (Fig. 6c right bottom; Supplementary Data 3) and found to be enriched for epithelial–mesenchymal transition (hypergeometric test, FDR < 0.05; Supplementary Table 6). Though the overlap of the microglia SDEGs with previous male and female microglia comparisons[33,41] was not significant, the overall result that female microglia are in a more immune-activated state was the same. Therefore, we conclude that the transcriptional differences between sexes appear in macrophages from the three tissues tested but that the specific differentially expressed genes are mostly tissue dependent.

**Comparison to human data**. To estimate the relevance of the murine transcriptional sexual dimorphism to humans, we studied the transcriptional sexual dimorphism identified in human monocytes (CD14 cells) and CD4 T cells of the ImmVar dataset, which collected immune cells from hundreds of healthy volunteers from three different ancestries (African-American, Asian, and Caucasian)[42–44]. Very few autosomal genes were identified as significantly different between sexes in the two populations with the smaller sample size, African-American and Asian. Thus we focused on the larger sample size population, Caucasian. In ImmVar Caucasians, 428 genes were differentially expressed between male and female CD4 T cells and 291 genes in monocytes (pFDR < 0.05). Of those, the mouse orthologs of 283 and 203 genes from the human CD4 T cells and monocytes,

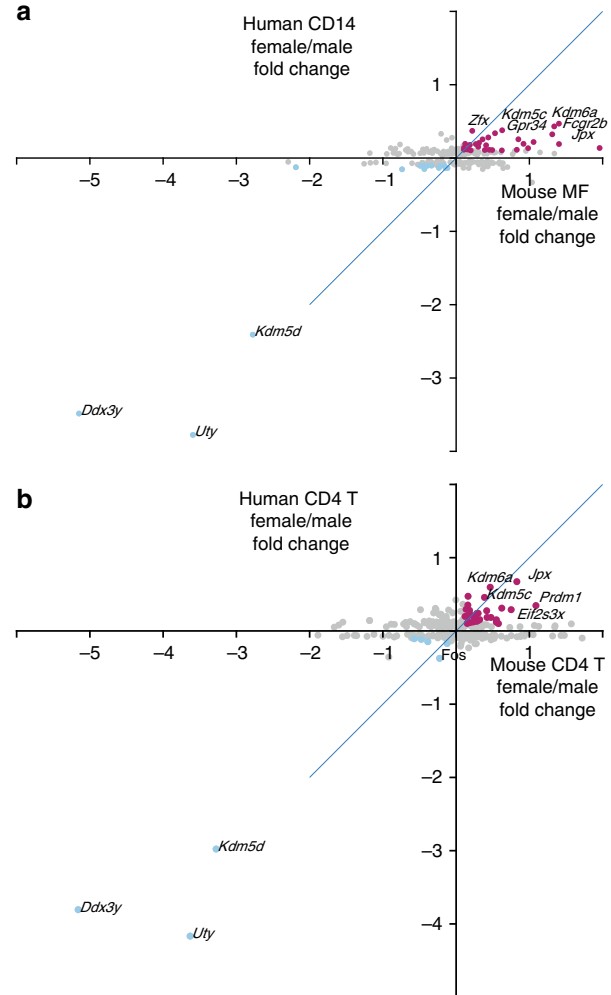

**Fig. 7** Comparison of human and mouse transcriptional immune sexual dimorphism. Female–male fold change (log₂) of ImmVar genes that are differentially expressed between female and male in **a** human CD14 and mouse MF cells from dataset B, and **b** human CD4 and mouse T4 cells from dataset B. The colored genes have >0.1 (log₂) fold change in both datasets in females (pink) or males (light blue). Selected gene symbols are shown. Source data for **a**, **b** are provided as a Source Data file

respectively, were expressed above noise threshold in murine T4 cells and MFs (dataset B, Supplementary Data 1 and 4). Of note, although significantly differentially expressed genes were identified in humans, their female to male fold change values were very low, a finding that matches the results reported in the current study. We compared the fold change of differentially expressed genes in ImmVar monocytes against our MFs (Fig. 7a, Supplementary Data 4) and ImmVar CD4 T cells against our CD4 T cells (Fig. 7b, Supplementary Data 4). Out of the 113 genes significantly higher in human female monocytes, 72 were also higher in murine female MFs (hypergeometric test $p$ 0.012). Looking at higher fold change thresholds, the human–mouse overlap becomes more significant—at 0.1 log₂(fold change) threshold, 50 human genes and 30 genes in common (hypergeometric test $p$ 0.003), and at 0.2, 14 human genes and 11 genes in common (hypergeometric test $p$ 0.0006). CD4 T cells presented a non-significant overlap between the human and murine SDEGs (hypergeometric test $p$ 0.079). These results reinforce the murine results, suggesting that MFs and possibly other innate immune cells present more sexually dimorphic gene expression than other immune cell types in mouse models as well as human subjects.

## Discussion

Sexual dimorphism in the mammalian immune system is manifested in many aspects, including the frequency and severity of infectious diseases and sex bias in autoimmune diseases and cancer prevalence[2,6,7]. To better understand the mechanism of the immune differences between males and females, we compared transcriptional profiles of unstimulated and IFN-stimulated immune cells and ATAC-seq from male and female mice. Phenotypic differences between male and female immunity exist regardless of any immune stimulation (e.g., baseline levels of activated T cells and immunoglobulins are higher in females[5,6]). We therefore expected to find transcriptional differences between males and females unstimulated immune cells.

Indeed, across the 11 immune lineages studied (pan-immune analysis), regardless of the extensive variability between the immune lineages, several female enriched pathways, which have previously been associated with known sexually dimorphic immune phenotypes, were identified. Activation of the IFNα and IFNγ responses pathway is highly correlated with autoimmune diseases, such as SLE[21,22], in which 90% of patients are female[8]. In addition, the complement and coagulation pathways play a role in autoimmune diseases, specifically in SLE[45,46]. The upregulation of genes that are part of those pathways in unstimulated female immunocytes may contribute to the imbalanced sex ratio of autoimmune diseases. The stronger and more robust inflammatory response in females[6] may be associated with the female-specific upregulation of the interleukin-6-JAK-STAT3 signaling pathway, which is involved in inflammatory response and autoimmunity[47].

It is possible that pan-immune differences between sexes are limited, as sexual dimorphism may be mediated by different genes in different cell types, or are limited, at the transcriptional level, to a few cell types. In light of these considerations, we also searched cell-type-specific sex differences. Surprisingly, only one cell type displayed differences between sexes, namely, the MF. We identified 26 SDEGs in female MFs; among these there were ISGs and genes of the complement system. Thus innate immunity, and specifically the IFN response and the complement system, may be involved in immune sexual dimorphism. This finding is in accordance with sex-specific expression quantitative trait loci and differential gene expression with small absolute effect identified in murine BMDM[32]. Among the female-specific SDEGs, we identified *Fcgr4* and *Fcgr2b*, which belong to the family of Fc receptors that binds the Fc portion of immunoglobulins[48]. The major function of *Fcgr4* is activation while *Fcgr2b* is an inhibitory receptor, and indeed, the activation of macrophages relies on this balance between activating and inhibiting immune complexes bearing Fc receptors[49]. The higher expression levels of Fc receptors in females might indicate different frequencies of activation states of unstimulated macrophages in females and males. Genes related to lipoprotein metabolism and the statin pathway were also upregulated in female MFs, in agreement with lipid metabolism exhibiting sexual dimorphism, mainly with respect to sex hormones[50]. It is possible that MFs from the region of the PC either affect or are affected by these differences. A possible explanation to the scarcity of the differences is that some differences are evident only at the protein level and not at the mRNA level, as shown for Toll-like receptor-4 and CD14, whose protein levels are higher in murine male macrophages compared to females, but their mRNA levels are the same[16,51].

SLE is the most drastic sex-biased autoimmune disorder with 90% female patients[8] and is associated with IFN pathway

changes[22]. Thus we chose to study the relevance of the transcriptional sexual dimorphism identified here in the context of SLE by comparing the female–male fold change of MF-ISGs with their SLE score, as calculated[35] on the PBMC expression profiles of two SLE studies[36,37]. PBMC include a variety of immune cell types, which are averaged in bulk expression profiles. Thus the comparison to the sorted MFs of the current study is limited. However, the absolute female–male fold change values from unstimulated macrophages are indeed higher in genes with greater association to SLE, suggesting possible involvement of macrophages and those specific genes in SLE.

IFN participates in signal transduction during infections and is involved in a variety of autoimmune disorders[21,52]. We used transcriptional profiling of the IFNα response in 11 immune cell types[35] (the same ImmGen 11 cell set that was used in this study) that was recently published to study the IFN-responsive genes in macrophages (MF-ISGs) in the context of male–female. Higher expression of MF-ISGs is observed in unstimulated MF females in comparison with males including significant bias in genes of the antiviral response. Even though both sexes exhibited extensive upregulation of MF-ISGs upon IFN stimulation, the gap between the sexes remained and the overall MF-ISG expression in stimulated female MFs was higher, not only for the genes of the antiviral module C3 but also for the genes of inflammation module C5, which became significant upon stimulation. It was previously noted that C3 regulators linked to their targets in both baseline state and after IFN stimulation, while C5 regulators linked to their target in the induced state only[35]. Therefore, baseline expression of IFN-related genes probably provides a stronger first line of defense for the female immune system (e.g., antiviral response) and after stimulation may contribute to the better outcome of females from infectious diseases by better activating immune pathways (such as inflammation).

To aid us in deciphering the regulation of the male–female transcriptional differences described above, we analyzed chromatin-accessible regions in three immune cell types in males and females. In accordance with the expression levels, genomic regions whose accessibility differed between males and females (DOs) were identified in MFs, but also in B and T4 cells. Both MFs and T4 cells presented more female DOs, including X-linked, specific functional regions. The most robust differentially accessible region was the X-linked locus carrying the long noncoding RNA *Firre* and an adjacent predicted gene, *Gm35612*. *Firre* escapes X-inactivation and is crucial for the maintenance of X-inactivation by anchoring the inactive X near the nucleus[39]. The sequence of *Firre* and local repeats in its locus were shown to diverge between rodents and other mammals, but its function seem to be conserved between human and mouse[53]. *Firre* organizes *trans*-chromosomal associations by recruiting specific loci located on different chromosomes to its own transcription site[54]. Sex-specific regulation of *Firre* in human CD4+ T cells is achieved by more active enhancers in females compared to that in males[55]. In addition, *Firre* was also recently shown to be controlled by NF-κB signaling and regulate the expression of several inflammatory genes in MFs through posttranscriptional mechanism[56]. Together with the regulatory role of *Firre*[54,56,57], our findings that *Firre* is more accessible in females MFs and CD4 T cells suggest that *Firre* might contribute to the sexually dimorphic gene expression.

As expected, *Xist*, responsible for the X-inactivation process[15], was more accessible in females. Additional X-linked OCRs (identified in both MF and T4 cells) are homologs to genes on the Y chromosome. Among these is *Kdm6a* (*Utx*), which is a H3K27me3 demethylase. *Kdm6a* is an homolog of the Y-linked gene *Uty* (T4 male TSS DO) and is related to the epigenetic mechanism of some immunological disorders, such as Kabuki

syndrome (KS). The largest subset of KS patients show higher susceptibility to infectious diseases and low serum immunoglobulin levels[58], which are characteristics associated with male immune phenotypes[5,6]. A smaller subset of KS displays autoimmune disorders, characteristic of the female immune phenotype[8], giving rise to the possibility that there may be mechanisms common to sexual immune dimorphism and KS subtypes.

As the only robust transcriptional sex effect in the ImmGen 11 cell set was identified in PC MFs, we extended our analysis to study tissue-resident MFs in two other tissues: Sp and CNS (microglia are considered the resident macrophages of the CNS). Several previous studies have demonstrated sexual dimorphism in innate immunity mainly at the phenotypic level[59]. However, to the best of our knowledge, at the transcriptomic level of immune cells, only male and female microglia have been compared, showing that female microglia are in a more immune-activated state[33,41]. The numbers of male–female differentially expressed genes found in splenic MF and microglia were even higher than those in PC-MFs, but the identity of those genes was mostly tissue specific. Our results suggest that sexual dimorphism characterize MFs in various tissues but is mediated by different genes in each tissue. Female MFs in both the PC and Sp present higher expression of immune pathways, with Sp-MFs presenting higher expression of the complement system, consistently with our pan-immune and cell-type-specific analyses.

Finally, we wondered how general our results are. On the one hand, even in the C57BL/6 strain we used, mice from substrains J and N display genetic and phenotypic differences, some of which are immune related[60]. On the other hand, stronger IFN response in female macrophages is conserved all the way to birds[12]. Thus we also analyzed the human ImmVar data from monocytes and CD4 T cells and show that, even though several hundred SDEGs are identified in human, the female–male fold change values are very low. The murine sample size (12–13 samples of each cell type) is too small to have the power to identify such small effects (Supplementary Note 1). Nonetheless, the overlap of SDEGs is significant between human monocytes and murine macrophages, indicating at least partial human–mouse conservation of transcriptional sexual dimorphism.

Differences in the frequency of cell types and sub-types between the sexes have been shown[28] and may contribute to differences in gene expression levels but are unlikely to mask differences in gene expression levels. Thus we do not assume that further stratifying cells into more coherent sub-types will reveal more differences between sexes in cell types where no differences were identified. Smaller effect sizes may be identified if much larger sample sizes were used. For macrophages, the only cell type for which SDEGs were identified, differences in frequencies of yet unidentified sub-types between sexes may underlie the sexual differential expression.

In summary, sexual transcriptional differences in the unstimulated immune system are mostly manifested in MFs and are evident in MF from the three tissues tested—PC-MFs, Sp-MFs, and microglia. Those differences increase in response to an IFN stimulus and are also reflected in the different levels of chromatin accessibility. Our results suggest that females might have an innate enhanced potential to withstand immune challenges due to more highly activated innate immune pathways prior to pathogen invasion. This female immune alertness, which makes females less vulnerable to infectious diseases, comes at the price of females being more prone to autoimmune diseases.

## Methods
**Mice.** C56BL/6J inbred mice were obtained from the ImmGen Colony at Jackson Laboratory, ME and housed in Harvard Medical School full barrier facility. We have complied with all relevant ethical regulations for animal testing and research.

Ethics oversight was by Harvard Medical School, under Institutional Animal Care and Use Committee protocol IS1257. Male and female mice of different ages were used as listed in Supplementary Data 1.

**RNA sequencing data generation**. Mice were sacrificed and immunocytes were isolated to high purity by flow cytometry according to the ImmGen SOP (https://www.immgen.org/Protocols/11Cells.pdf). For stimulated and unstimulated mice, Sp, whole CNS, and PC were harvested.

Eleven immune cell types from male and female mice were profiled by RNA-seq. This 11 cell set is currently the standard adopted by the Immunological Genome Project (ImmGen[61]) for the study of infectious and immunologic challenges; it encompasses all the major immunocyte lineages, namely, GNs, DCs, MFs, B, T4 and T8, Treg cells, NK and NKT cells, and Tgd cells. MFs and B1a cells were sorted from the PC and all other cell types were sorted from the Sp. Antibodies from eBioscience 48–0193, 53–1021, 12–0431, 25–4801, 17–0051, 48–5961, 11–5890, 12–0621, 25–0193, 17–0081, 47–0042, 48–5961, 48–0452-82, 11–5321, 25–4801, 56–0114-82, 12–1351-82, 17–5931-82, 48–5961, 11–0042, 12–5941, 46–5711, and 17–0251 were used (See Supplementary Fig. 8 and Supplementary Data 1 for gating strategies, antibody dilutions, surface markers, and more details).

After the final sort of 1000 cells directly into 5 µl lysis buffer [TCL Buffer (Qiagen) with 1% 2-Mercaptoethanol], Smart-seq2 libraries were prepared following Picelli et al.[62,63], with slight modifications. Briefly, total RNA was captured and purified on RNAClean XP beads (Beckman Coulter). Polyadenylated mRNA was then selected using an anchored oligo(dT) primer (5′-AAGCAGTGGTATCAACGCAGAGTACT30VN-3′) and converted to cDNA via reverse transcription. First-strand cDNA was subjected to limited PCR amplification followed by Tn5 transposon-based fragmentation using the Nextera XT DNA Library Preparation Kit (Illumina). Samples were then PCR amplified for 18 cycles using barcoded primers such that each sample carries a specific combination of eight-base Illumina P5 and P7 barcodes and pooled together prior to Smart sequencing. Smart-seq paired-end sequencing was performed on an Illumina NextSeq500 using 2 × 25 bp reads with no further trimming.

Data were collected in four independent experiments for a total of 92 female samples and 91 male samples, described below (Fig. 1a, Supplementary Data 1):

Dataset A (11-cell-set ages) included 2–4 repeats, each pooled from 3 mice, for 3 different ages (one repeat from each age, young—2 months; adult—6 months; and old—17/20 months in females/males, respectively) for 11 unstimulated immune cell types from males and females (66 samples in total, Supplementary Data 1).

Dataset B (11-cell-set NVE-IFN) included 3 repeats, each pooled from 3 mice, for 6-week-old male and female of 11 unstimulated (NVE) immune cell types from males and females (66 samples) and 24 samples of 3 immune cell types (B, GN, MF) after stimulation by type-1 IFN, (90 samples in total, Supplementary Data 1). For stimulation, 1000 (1k) or 10,000 (10k) enzyme units of IFNα (R&D SYSTEMS Catalog #12100-1) were delivered to 12 mice (3 males and 3 females for each dose) via subcutaneous injection, and mice were sacrificed 2 h later. The 24 stimulated samples comprised of male and female triplicates for 1k IFN (B cells only) and 10k IFN (B, GN and MF).

Dataset C (tissue MFs) comprised 2–3 replicates, each taken from single mouse, for unstimulated MFs originating from the PC, the Sp, or the CNS, from 6-week (PC and Sp) and 8-week (CNS) old male and female mice. While PC and Sp originated from the same pool of mice, CNS samples were produced from independent mice pool (15 samples in total, Supplementary Data 1).

Dataset D (ATAC-seq matched mRNA) included 2 repeats, each taken from single mouse, of 3 (of 11 cell set) cell types, namely, MF (PC), B, and T4 cells (Sp), of ultra-low input RNA-seq of the ATAC-seq samples (12 samples in total, Supplementary Data 1).

**RNA-seq pre-processing**. For datasets A and B, reads were mapped to the mouse genome (mm10) using hisat[64] (0.1.6-beta release). Bam files were sorted and indexed by SAMtools[65] (1.2 release). Assembly, quantification, and normalization were performed using CuffLinks (1.2 release), according to the Tuxedo pipeline[66]. A merged transcriptome constructed from all samples of all datasets was used as a reference annotation for quantification (by CuffQuant) and normalization (by CuffNorm).

For dataset C, reads were aligned to the mouse genome (GENCODE GRCm38/mm10 primary assembly and gene annotations vM16; https://www.gencodegenes.org/mouse_releases/16.html) with STAR 2.5.4a (https://github.com/alexdobin/STAR/releases). The ribosomal RNA gene annotations were removed from GTF (General Transfer Format) file. The gene-level quantification was calculated by featureCounts (http://subread.sourceforge.net/). Raw read count tables were normalized by median of ratios method with DESeq2 package from Bioconductor (https://bioconductor.org/packages/release/bioc/html/DESeq2.html).

For dataset D, low-input RNA-seq data of three unstimulated cell types, MF, B, and T4 cells (with two replicates), from male and female mice were taken from the recent ImmGen dataset of ATAC-seq[38]. The normalized fragment per kilobase of exon per million reads mapped (FPKM) values of the RNA-seq samples (matched to those of ATAC-seq data) were used in the current study.

For all datasets, normalized FPKM values that were lower than one were replaced by one, and all values were $\log_2$ transformed. To reduce noise and maximize power, only genes with a $\log_2$ expression higher than noise threshold (five in datasets A, B, and D; six in dataset C) in at least as many samples as the size of smallest group (six samples in datasets A and B, and two in datasets C and D) were included in all downstream analyses. For PCA analyses, correlation calculations, and heatmap presentations, the $\log_2$-transformed FPKM expression values of the genes that passed the noise threshold were normalized by subtraction of the gene mean expression, followed by division by the standard deviation.

**Data quality control**. To evaluate the quality of the data and to remove low-quality samples, Fastqc (https://www.bioinformatics.babraham.ac.uk/projects/fastqc/) and the stats option of SAMtools[65] were used. Samples with less than one million mapped and paired reads were not included in the downstream analysis. Of the 196 samples produced, 13 were discarded owing to low quality (4 from dataset A, 6 from IFN-induced male and female T4 cells from dataset B, and 3 from the tissue MF dataset C), leaving 183 samples for downstream analysis (Supplementary Data 1).

**Principal component analysis**. Normalized expression values were used as input to the Matlab function pca. Plots show the specified principal components. Genes were considered as contributing to a principal component if the absolute value of their coefficient for this component was >0.05.

**Pan-immune differential expression**. To identify pan-immune differences between males and females, samples from all 11 cell types were subjected to a two-sided paired $t$ test (pairing cell types and datasets, biological replicates in dataset B were randomly paired). For dataset A, the corresponding age samples of the samples that had been removed during quality control were also excluded.

Pan-immune sex-biased genes were defined as genes that conform with the following two criteria in expression level analysis: (1) At least a 1.5-fold change difference between male and female [|$\log_2$ fold change mean(female)/mean(male)| $>\log_2(1.5)$], and (2) statistically significant differential expression according to a $t$ test followed by the Benjamini–Hochberg FDR procedure[67] for multiple comparison problems. FDR-adjusted $p$ values (pFDR) were calculated by the MATLAB function mafdr, and the significance threshold was set to pFDR < 0.2. Based on the resulting $t$-statistic, a Pre-ranked Gene-Set Enrichment Analysis (GSEAPreranked) was applied using Hallmark gene sets from MsigDB[34,68].

**Cell-type-specific differential expression**. Cell-type-specific SDEGs were defined as genes that conform with the following two criteria in the expression level analysis: (1) statistically significant differential expression according to a two-sided paired $t$ test (applied per cell type on datasets A and B together. Samples were paired by dataset and age. Biological replicates in dataset B were randomly paired. pFDR < 0.2), and (2) at least a 1.5-fold change difference between male and female expression levels [|$\log_2$ fold change mean(female)/mean(male)| $>\log_2(1.5)$]. The same analysis was applied to identify differential expression in the IFN analysis, using an unpaired $t$ test (applied only on dataset B) with more stringent cutoffs (pFDR < 0.05 and at least a two-fold change), as the IFN effect is stronger than the sex effect.

**Permutation test**. To estimate the reliability of identified SDEGs, bootstrap permutations were performed 1000 times. Two types of designs were used: (1) Keeping the male–female pairing as in the experiment (to avoid confounding effects of cell type, dataset, or age) and randomly assigning male or female labels (used for all paired $t$ tests, pan-immune and cell-type-specific differential gene expression). (2) Randomly assigning male or female labels based on group size (used for all differential gene expression in the IFN analysis). For each $t$ test, both $t$ test $p$ value and permutation test $p$ value are reported.

**ISG signature**. Mostafavi et al. profiled the transcriptome of IFN response in ImmGen's 11 cell set and defined ISGs as genes upregulated in at least one cell type with >2-fold induction and pFDR < 0.1 (975 genes)[35]. Of those, 601 that were significantly upregulated in MF after IFN stimulation were termed MF-ISGs. Only the MF-ISGs that were expressed above the noise threshold in at least 3 samples were included in our analyses (71 in the unstimulated MF only and 313 including the IFN samples).

**Two-way ANOVA for tissue macrophages**. To identify the differences between males and females in MFs within and across the PC, Sp, and CNS (microglia), a two-way ANOVA followed by a multiple comparison procedure was performed using the Matlab functions anovan and multcompare (dataset C, Supplementary Data 1); the variables were the overall sex effect, the tissue effect, and the effect of sex and tissue interaction. To consider a gene as contributing to an effect, it must conform to both of the following criteria: (1) statistically significant two-way ANOVA $p$ value for the effect (pFDR < 0.2); and (2) at least a 1.5-fold change in the comparisons relevant for the effect, namely: for testing the sex effect, male–female comparison in all three tissues; for testing the sex–tissue interaction effect, at least one male–female comparison. In the case of sex–tissue interaction effect, genes

with tissue-specific differential expression were required to express above threshold in at least all male or female samples of the specific tissue.

To evaluate the resulting gene lists, a functional enrichment analysis was performed. Gene sets were downloaded from the Molecular Signature Database (MsigDB) using Hallmark gene sets[34,68], and enrichment for functional annotations was calculated by using the hypergeometric test (Matlab function hygecdf). The number of background genes was set to 10,264 (genes expressed above the noise threshold) for the MF tissue analysis. The Benjamini–Hochberg FDR procedure[67] was applied for multiple testing correction (pFDR < 0.05).

**ATAC-seq data**. Samples of three unstimulated cell types, MF, B, and T4 cells (with two replicates), from male and female mice were previously profiled[38]. Matched ATAC-seq and RNA-seq (here termed dataset D) from the same batches of sorted cells were generated by splitting for expression profiling by low-input RNA-seq and for chromatin accessibility analysis by fast-ATAC-seq[69]. The OCRs activity index was normalized across all cell-types by quantile normalization and filtered following Yoshida et al.[38]. In the current study, the definition of OCRs and their association with genes was adopted from Yoshida et al.[38] as well as the parsing of OCRs to the TSS or DEs.

**Differential OCRs**. Normalized OCR expression data were analyzed on a $\log_2$ scale, and mean peak values were used for each population. To estimate the significance of the difference, we used an adjusted $t$ test following Zeisel et al.[70]. Under the assumption that most genes are not differentially expressed, all genes can be used to estimate the distribution and calculate the $p$ value, for specific genes based on this estimation. For each OCR, we used the distribution of all peaks with similar value (one unit window after $\log_2$), based on their median absolute deviation (MAD) instead of mean, as MAD is more robust than mean and will not be affected by few differentially expressed OCRs. The code is available under (https://doi.org/10.5281/zenodo.3377594) function CompareTwoChipsV2.m.

OCRs that conformed to the following criteria were identified as DOs: (1) A significant differential peak value between sexes (adjusted two-sided $t$ test pFDR < 0.05); and (2) at least a 2-fold change difference between male and female mean peak values.

**ImmVar probe filtering and expression quantification**. Raw microarray expression files were processed separately for each cell type using the oligo R package (https://www.bioconductor.org/packages/release/bioc/html/oligo.html). Robust Multichip Average background correction and quantile normalization were performed on the whole set of probes ($N = 1,102,500$ 25-mer features).

Probe sequences were mapped to the NCBI *Homo sapiens* GRCh38 assembly with bowtie2[71] using a global alignment strategy (penalty score = −15.6) allowing up to 5 alignments per read. Probes overlapping GENCODE v.22 gene exons were identified using bedtools intersect[72].

These mapping information were used to identify and remove probes, which fell into one of the following three categories:

1. Overlapped a single-nucleotide polymorphism at minor allele frequency >0.1 in any of the EAS, CAU, or AFR 1000 Genomes populations ($N = 39,349$ probes)
2. Multimapped exons at different genomic positions ($N = 50,437$ probes)
3. Targeted Y-linked genes outside of the pseudo-autosomal region but exhibited high expression levels in females ($N = 496$ probes)

Affymetrix annotations specify 764,886 gene-mapping probes; however, 58,238 probes did not map to any GENCODE v.22 exon. These probes, and the 83,566 probes flagged in at least one of the above steps, were removed, resulting in a set of 623,081 probes used for downstream analysis.

Probe expression values were summarized at a gene level, using the empirical probe-exon annotations from the mapping step, via the median-polish algorithm implemented in oligo. Genes supported by <5 probes or which had >75% of probes removed were excluded. Within each cell type, genes expressed below the 10% quantile in >2/3 of males and >2/3 of females were flagged for removal.

In CD4+ T cells, 21,358 genes passed these filters. In CD14+ cells, 21,310 genes passed this filtering. Twenty-one thousand and eighty-six genes passed filtering in both cell types. In total, the 21,582 unique genes that passed filtering in at least one cell type were included in downstream analyses.

**ImmVar differential expression analysis**. For each gene, the effect of sex on expression was calculated using the following linear model (Eq. 1):

$$y_g \sim \beta_{\text{sex}} + \sum_i \beta_i SV_i \qquad (1)$$

where $y_g$ is a vector of expression for the $g$th gene, sex is a vector coding male as 1 and female as 0, and accounting for $i$ surrogate variables estimated by the Buja & Eyuboglu (be) method[73] implemented in the sva R package (version 3.18.0 http://bioconductor.jp/packages/3.2/bioc/html/sva.html).

To test for differences in the mean gene expression levels between males and females, differential expression analysis was performed separately for each cell type using the empirical Bayes (eBayes) function of the limma R package. Significance

was deemed at FDR[67] < 0.05. The π1 statistic[74] for each $p$ value distribution were calculated using the Bioconductor's $q$ value R package (version 1.36 https://bioconductor.org/packages/2.13/bioc/html/qvalue.html).

**Comparing human and mouse**. To compare between human and mouse differentially expressed genes, all human gene symbols were translated to their aliases in mouse (using ENSEMBL BioMart data mining-tool[75] and compared to the genes in dataset B. ImmVar's monocytes (CD14) and CD4 T cells were compared to ImmGen's macrophages and T4 cells, respectively. Only genes that were differentially expressed in ImmVar's Caucasian population (q-FDR < 0.05, 428 and 291 genes in CD4 and monocytes, respectively) and expressed >0 in at least three of the samples in ImmGen's compared cell type were used, leaving 283 and 203 genes in CD4 and monocytes, respectively (dataset B, Supplementary Data 4). To evaluate the significance of consistent fold change direction between mouse and human, hypergeometric test was used (Matlab function hygecdf).

**Reporting summary**. Further information on research design is available in the Nature Research Reporting Summary linked to this article.

## Data availability

All RNA-seq data sets generated in this manuscript have been deposited in the GEO under accession number GSE124829. Samples of dataset C are also part of GSE122108. ATAC-seq samples are part of GSE100738. Human ImmVar data accession number is GSE56035. The source data underlying all Figures and Supplementary Figures are provided as a Source Data file. All other data are included in the supplemental information or available from the authors upon reasonable request.

## Code availability

The code used to identify differential OCRs is publicly available at https://doi.org/10.5281/zenodo.3377594.

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

## Acknowledgements

The authors wish to thank the ImmGen consortium, especially Professor Christophe Benoist. Research reported in this publication was supported by Broad-Israel Science Foundation Grant 1644/15, Israel Science Foundation Grant 500/15, and the National Institute of Allergy and Infectious Diseases of the National Institutes of Health under Award Number R24AI072073. The content is solely the responsibility of the authors and does not necessarily represent the official views of the National Institutes of Health. S.T. G.-O. was supported by Hi-Tech, Bio-Tech, and Chemo-tech fellowship of Ben-Gurion University of the Negev.

## Author contributions

K.S., H.Y., and B.M. performed the experiments. C.C. and B.E.S. performed the ImmVar analysis. O.Z. performed the power analysis. S.T.G.-O. performed all other analyses, with help from H.N.-G. and N.E. T.S. supervised the research. All authors contributed to manuscript writing and revision.

## Competing interests

The authors declare no competing interests.
