## [Peer Review File · Nature Communications]

Reviewers' comments:

Reviewer #1 (RNAseq, systems biology)(Remarks to the Author):

SUMMARY

It is well-established that females and males have different immune responses, yet studies addressing the underlying causes at scale are lacking. Gal-Oz et al., set out to determine the differences between male and female immune cell types and take a gene-expression based approach. Surprisingly, they find that across 11 different cell populations, only one cell type, MFs, displays sex-specific differences in gene expression. The 41 gene cohort they identify in MFs shows a female-specific enrichment in IFN-stimulated genes. And the authors show that these genes are found at higher basal level expression, and suggest that these findings lead to female immune alertness. The central finding in this paper that there is a group of sex-specific genes in MFs that could help explain immune differences in males and females. This is a comprehensive study that provides a key insight that, as measured by RNA, the immune system is not dramatically sexually dimorphic with the exception of Macrophages. This study will be of great interest of the Nature communications readership. Below are the following considerations for the authors.

MAJOR POINTS:

1. The core of the paper is the identification of the 41 SDEGs in MFs and this is derived from RNA-seq datasets from young, adult, and aged mice. It has been shown that gene expression can change in the immune populations from young and aged mice. The authors should comment on the distribution of the 41 SDEGs relative to young, adult, and aged individual analyses. For example, are the 41 SDEGs present throughout life (young, adult, and aged analyses) or do they arise at a specific age?
2. The 41 SDEGs are central to the paper. It would help to visualize these differences by plotting FPKM values for a cohort of individual genes from males and females. That way a reader can see the abundance as well as difference.
3. The authors should comment on what is known about the sex-specific regulation in macrophages (gene and physiological level) in order to provide their findings more of a context. For examples: (1) TLR4 (innate immune response receptor) is more highly expressed on male macrophages. (2) In chickens, female macrophages the interferon response gene network is set at a higher transcriptional level than males.

Sex-specific regulation of immune responses by PPARs

PMID: 28775365

Sex Specific Gene Regulation and Expression QTLs in Mouse Macrophages from a Strain Intercross

PMID: 18197246

Cell-Autonomous Sex Differences in Gene Expression in Chicken Bone Marrow-Derived Macrophages

PMID: 25637020

4. Since the central finding of this paper is that that there is a group of sex-specific genes in MFs that could help explain immune differences in males and females, yet the context of the 41 SDEG is unclear as related to innate immunity / immune function. The authors should curate a table of the SDEGs in MFs and indicate any known mouse phenotypes and/or human disease in order to put these findings into more of a context.

5. The methods in this manuscript are poorly described. With this impressive number of samples and that IMMGEN aims to provide a standard for the field, it is essential to describe the experimental setup that went into each analysis:

(a) The RNA-seq experiment design is not described clearly in the text, figure 1A, or the methods. For example, Figure 1A shows the clustering for 11 different cell types. There is a lot of information that is unknown such as: (1) Per sample, how many replicates are female? (2) How many replicates are male? (3) Are the RNA-seq samples derived from pooled mice? If so, how mice many per pool? (4) The methods section indicates that "young," "adult," and "aged" mice were used. How many replicates are from each age group?

(b) In the pan-immune approach, that identified 14 SDEGs, it is unclear what comparison was used in order to identify this group of 14 genes. This is in part because from Figure 1A and from the methods section, it is also unclear what exactly comprises dataset A and B and how they differ. "The pan-immune approach was implemented by applying a paired t-test to the RNA sequencing datasets described above (datasets A and B, Fig. 1a), in which samples were matched by cell type, age and dataset."

(c) The flow cytometry methods section is missing. What surface markers were used to sort each cell population?

(d) The methods describing the in vivo INFalpha experiment are unclear and missing detail. This comment specifically refers to the part: "we analyzed RNA-seq profiles of three cell types, GN, MF and B cells, from male and female mice before and after exposure to IFNa." From the methods section, which describes: "For stimulation, mouse were intraperitoneal injected with 1k and 10k Interferon alpha corresponding to 1,000 and 10,000 enzyme units and sacrificed two hours later." What is 1k and 10k referring too? What was the manufacture of INFalpha? How many mice were injected? Males, females, age? The authors must include this information in a clear manner.

(e) The authors must clarify what strain of C57BL6 was used J or N (or a mix) since there are reported differences in blood cells between J and N strain and because IMMGEN is a standard for the field. The methods only lists "C57BL6."

Elevated Oxidative Stress Impairs Hematopoietic Progenitor Function in C57BL/6 Substrains

PMID: 30017822

6. This paper involves mouse experiments, but there is no mouse ethic statement, no statement on what type of facility the mice were housed in (pathogen-free, pathogen-specific, etc), and there is no institutional oversight statement from IACUC in the methods section. These statements must be included.

7. The authors should put their findings into the context of what is known about sexual dimorphic gene expression in mammals. This area is of interest because there appears to be conflicting studies - some show mammals have dramatic differences in gene expression between males and females and others do not. This study is one of the more complete and modern and of the opinion that this finding alone is very important. Here are other studies that started to debate "how sexually dimorphic" gene-expression:

Sexual dimorphism in mammalian gene expression.

PMID: 15851067

Major molecular differences between mammalian sexes are involved in drug metabolism and renal function.

PMID: 15177028

Sex-Specific Selection and Sex-Biased Gene Expression in Humans and Flies

<https://journals.plos.org/plosgenetics/article?id=10.1371/journal.pgen.1006170>

Tissue-specific expression and regulation of sexually dimorphic genes in mice

PMID: 16825664

8. In this study, they re-analyze existing ATAC-seq data (IMMGEM) and point out that there are some differences in open chromatin regions between males and females – Firre, GM35612, and Pls3 (With respect to Fig.5C,D). The authors should put this finding into more of a biological context. For example (1) previous reports find that Firre escapes X inactivation and there are regulatory differences in the inactive and active X. This was determined by allele specific chip and RNA seq.

Function and evolution of local repeats in the Firre locus

PMID: 27009974

Topological organization of multichromosomal regions by the long intergenic noncoding RNA Firre

PMID: 24463464

Regulation of the ESC transcriptome by nuclear long noncoding RNAs

PMID: 26048247

The lncRNA Firre anchors the inactive X chromosome to the nucleolus by binding CTCF and maintains H3K27me3 methylation

PMID: 25887447

MINOR POINTS:

1. What do the 5 different modules represent? It is not clearly stated. “The highly complicated interferon response of the same cell types as here was recently parsed into five modules, that are different in terms of the cells in which they are active, their regulation, their tonic interferon response, and more.”

2. Missing Citation: “An example of this potential effect is the X-linked gene DEAD-box helicase 3 (Ddx3x) and its Y-linked homolog (Ddx3y). Ddx3x is crucial for interferon (IFN) production in response to pathogens, and high levels of Ddx3x can boost the female IFN inducers response.”

Reviewer #2 (Immune ageing, systems immunology)(Remarks to the Author):

The manuscript addresses the highly relevant question of sexual dimorphism in immune responses and autoimmune. The approach taken to extend the transcriptome data base of the Immunological

Genome Project to include female mice and extend this study to include ATAC-seq data and transcriptional data after in vivo challenge is timely. Data generation followed standard operating procedure developed by ImmunGen and quality control and analytical tools appear to be overall appropriate. The most striking finding is that gender differences were small, of borderline significance and not consistent across tissue, even not related myeloid tissues.

1. Given the small differences, a discussion on the power of the study would be desirable.
2. Given that gender signatures are tissue specific, the potential limitation that cells were not stratified for differentiation status should be discussed. For example, there are substantial differences between naïve and memory T cells in transcriptomes and chromatin accessibility maps.
3. The study included three different age groups, but age is only mentioned in passing in the result section. If age is important variable, could the inclusion of three different age groups reduced the powered?
4. What was the rationale for using different FDRs for different comparisons?
5. Although of interest, the comparison to SLE transcriptome analysis remains superficial. It is unclear whether the two chosen comparison datasets are representative. There is no discussion on the heterogeneity of cell types in PBMC that account for the SLE signature in the chosen datasets and the SLE score is not well explained.
6. The findings on Firre are of interest, but preliminary and should not be overinterpreted. It is certainly not justified to make Firre a focus point in the abstract.
7. The data appear to be consistent with the interpretation that female cells are less able to maintain quiescence and macrophages are the more responsive cell type. Are the observed signatures truly an IFN response signature or do they just reflect macrophage activation?
8. The authors should discuss that the human studies but not the mouse studies show a signature for T cells. The manuscript by Howard Chung's group in Cell Systems that ATAC-seq signatures in PBMCs are driven by gender should be discussed.

Minor comments

Incomplete sentence in the introduction (Landscape of X chromosome inactivation across human tissues).

Reviewers' comments:

Reviewer #1 (RNAseq, systems biology)(Remarks to the Author):

SUMMARY

It is well-established that females and males have different immune responses, yet studies addressing the underlying causes at scale are lacking. Gal-Oz et al., set out to determine the differences between male and female immune cell types and take a gene-expression based approach. Surprisingly, they find that across 11 different cell populations, only one cell type, MFs, displays sex-specific differences in gene expression. The 41 gene cohort they identify in MFs shows a female-specific enrichment in IFN-stimulated genes. And the authors show that these genes are found at higher basal level expression, and suggest that these findings lead to female immune alertness. The central finding in this paper that there is a group of sex-specific genes in MFs that could help explain immune differences in males and females. This is a comprehensive study that provides a key insight that, as measured by RNA, the immune system is not dramatically sexually dimorphic with the exception of Macrophages. This study will be of great interest of the Nature communications readership. Below are the following considerations for the authors.

MAJOR POINTS:

1. The core of the paper is the identification of the 41 SDEGs in MFs and this is derived from RNA-seq datasets from young, adult, and aged mice. It has been shown that gene expression can change in the immune populations from young and aged mice. The authors should comment on the distribution of the 41 SDEGs relative to young, adult, and aged individual analyses. For example, are the 41 SDEGs present throughout life (young, adult, and aged analyses) or do they arise at a specific age?

We acknowledge this point should be clarified, and clarified accordingly in the results section 'Cell-type-specific sex signature' (p.8). In both pan immune and cell type specific tests, we paired the samples by age, as now clearly stated in text. We added a new supplementary figure 2 of the SDEGs heatmaps with colorbars indicating sex, age and dataset to address this point (Suppl. Figure 2A). As Figure 1B below shows, the standard deviation of the 41 SDEGs in the three samples that are of the same age is mostly similar to that in the three samples from different ages for males. For females (Figure 1A below), the old female is a bit of an outlier (as can be seen in paper Figure 3b and Suppl. Figure 2A) – not enough to be removed, but enough to drive the standard deviation of 5 SDEGs to be higher in females. We also show below, two figures (Figure 1C-D) with the patterns of all 41 SDEGs (separately for male and female SDEGs). Both show that the 41 SDEGs are differentially expressed between male and female, independent of age.

However, as the reviewers predicted, some genes seem to be more different in specific ages. Unfortunately, our design cannot robustly identify such effects, but we hope to further study them in future studies.

Figure R1. Age effect on SDEGs. **Comparison of standard deviation (STD)** of the samples in Dataset A (different ages, Y axis) and Dataset B (same age, X axis), in **(A)** females and **(B)** males. **(C-D)** **Expression patterns** of all **(C)** female specific and **(D)** male specific MF SDEGs across ages, from Datasets A (different ages, circles) and Dataset B (same age, triangles) in males (light blue) and female (pink) samples.

2. The 41 SDEGs are central to the paper. It would help to visualize these differences by plotting FPKM values for a cohort of individual genes from males and females. That way a reader can see the abundance as well as difference.

We add a non-normalized heatmap of the 41 SDEGs FPKMs as Suppl. figure 2B. These can also be seen in the SDEGs pattern Figure R1 C-D above.

3. The authors should comment on what is known about the sex-specific regulation in macrophages (gene and physiological level) in order to provide their findings more of a context. For examples: (1) TLR4 (innate immune response receptor) is more highly expressed on male macrophages. (2) In chickens, female macrophages the interferon response gene network is set at a higher transcriptional level than males.

Sex-specific regulation of immune responses by PPARs
PMID: 28775365

Sex Specific Gene Regulation and Expression QTLs in Mouse Macrophages from a Strain Intercross
PMID: 18197246

Cell-Autonomous Sex Differences in Gene Expression in Chicken Bone Marrow–Derived Macrophages
PMID: 25637020

We thank the reviewer for these excellent and relevant references, and added them to the introduction (P.5) and discussion (P.15-16).

Regarding TLR4, the protein expression is indeed higher in resting male macrophages compared to females, (as reviewed in PMID: 28775365), but there is no difference at the mRNA level, both in the original paper (PMID: 16574244) and in our data. We added this in the discussion, as one possible explanation to the scarcity of the differences we see.

4. Since the central finding of this paper is that there is a group of sex-specific genes in MFs that could help explain immune differences in males and females, yet the context of the 41 SDEG is unclear as related to innate immunity / immune function. The authors should curate a table of the SDEGs in MFs and indicate any known mouse phenotypes and/or human disease in order to put these findings into more of a context.

We added the suggested information as a Supplementary Note (Supplementary Note 2), by a comprehensive search of immune related phenotypes of each gene in PubMed, OMIM, gene cards and JAX. We have also added a discussion about two FCGRs that are expressed higher in females (discussion P.16).

5. The methods in this manuscript are poorly described. With this impressive number of samples and that IMMGEN aims to provide a standard for the field, it is essential to describe the experimental setup that went into each analysis:

(a) *The RNA-seq experiment design is not described clearly in the text, figure 1A, or the methods. For example, Figure 1A shows the clustering for 11 different cell types. There is a lot of information that is unknown such as: (1) Per sample, how many replicates are female? (2) How many replicates are male? (3) Are the RNA-seq samples derived from pooled mice? If so, how mice many per pool? (4) The methods section indicates that “young,” “adult,” and “aged” mice were used. How many replicates are from each age group?*

All samples are detailed in supplementary table 1, including the number of samples of each sex and age in each cell type. We added a clear reference in the text (result section Transcriptional profiling, p.5-6).

This number of samples per age is now added to Methods section- RNA sequencing data, p.19.

We apologize for omitting the pooling information. This information is now added to Method section RNA sequencing data generation (P.20-21). Samples from datasets A and B were pooled from 3 mice while samples from datasets C and D were taken from a single mouse.

(b) *In the pan-immune approach, that identified 14 SDEGs, it is unclear what comparison was used in order to identify this group of 14 genes. This is in part because from Figure 1A and from the methods section, it is also unclear what exactly comprises dataset A and B and how they differ. “The pan-immune approach was implemented by applying a paired t-test to the RNA sequencing datasets described above (datasets A and B, Fig. 1a), in which samples were matched by cell type, age and dataset.”*

We clarified the text result section- Pan-immune sex signature, p.6.

“ Datasets A and B were produced independently, and are different in the ages of the samples, the protocols and the depth of sequencing. The pairing was done by dataset and age, to account for batch effects and age related changes, respectively. Though age related changes could be sex dependent, the number of samples in each age group does not allow identification of such effect accurately.”

(c) *The flow cytometry methods section is missing. What surface markers were used to sort each cell population?*

We apologize for omitting this information. The surface markers used for each population have been added to Supplementary table 1, as well as the antibodies

used. We also added relevant references to the table in method section: "RNA sequencing data generation".

Data was generated according to the ImmGen SOP. To clarify this point we added:

"Mice were sacrificed and immunocytes were isolated to high purity by flow cytometry according to the ImmGen SOP (<https://www.immgen.org/>). For stimulated and unstimulated mice, spleen, whole CNS and peritoneal cavity were harvested."

(d) The methods describing the in vivo INFalpha experiment are unclear and missing detail. This comment specifically refers to the part: "we analyzed RNA-seq profiles of three cell types, GN, MF and B cells, from male and female mice before and after exposure to IFNα." From the methods section, which describes: "For stimulation, mouse were intraperitoneal injected with 1k and 10k Interferon alpha corresponding to 1,000 and 10,000 enzyme units and sacrificed two hours later." What is 1k and 10k referring too? What was the manufacture of INFalpha? How many mice were injected? Males, females, age? The authors must include this information in a clear manner.

We apologize for omitting this information. IFNa manufacturer is R&D SYSTEMS Catalog #12100-1. We have organized the section and added the missing information to the Methods section "RNA sequencing data".

"Dataset B ("11-cell-set" NVE-IFN) included 3 repeats for 6 weeks old male and female of 11 unstimulated (NVE) immune cell types from males and females (66 samples in total) and 24 samples of 3 immune cell types (B, GN, MF) after stimulation by type-1 interferon (IFN), (90 samples in total, Supplementary Table 1). For stimulation, 12 mice were intraperitoneal injected with IFN-alpha (R&D SYSTEMS Catalog #12100-1), 3 males and 3 females were injected with 1,000 (1k) or 10,000 (10k) enzyme units, and mice were sacrificed two hours later. The 24 stimulated samples comprised of male and female triplicates for 1k IFN (B cells only) and 10k IFN (B, GN and MF)."

(e) The authors must clarify what strain of C57BL6 was used J or N (or a mix) since there are reported differences in blood cells between J and N strain and because IMMGEN is a standard for the field. The methods only lists "C57BL6."

We apologize for omitting this information. We added a "Mice" paragraph to the method section, which includes the mice strain.

"C56Bl/6 J inbred mice were obtained from the ImmGen Colony at Jackson Laboratory, Maine"

Elevated Oxidative Stress Impairs Hematopoietic Progenitor Function in C57BL/6 Substrains

PMID: 30017822

We thank the reviewer for pointing out the difference between the B6 substrains, and add the substrains difference and the suggested reference (as well as two others showing stroke outcome and neurological performance differences between strains) and another one to the discussion on page 18.

6. This paper involves mouse experiments, but there is no mouse ethic statement, no statement on what type of facility the mice were housed in (pathogen-free, pathogen-specific, etc), and there is no institutional oversight statement from IACUC in the methods section. These statements must be included.

We apologize for omitting this information. In the newly added "Mice" paragraph we added housing and IACUC protocol.

" C56Bl/6 J inbred mice were obtained from the ImmGen Colony at Jackson Laboratory, Maine, and housed in a full barrier facility. Ethics oversight by Harvard Medical School, under Institutional Animal Care and Use Committee protocol IS1257."

7. The authors should put their findings into the context of what is known about sexual dimorphic gene expression in mammals. This area is of interest because there appears to be conflicting studies -- some show mammals have dramatic differences in gene expression between males and females and others do not. This study is one of the more complete and modern and of the opinion that this finding alone is very important. Here are other studies that started to debate "how sexually dimorphic" gene-expression:

Sexual dimorphism in mammalian gene expression.
PMID: 15851067

Major molecular differences between mammalian sexes are involved in drug metabolism and renal function.
PMID: 15177028

Sex-Specific Selection and Sex-Biased Gene Expression in Humans and Flies
<https://journals.plos.org/plosgenetics/article?id=10.1371/journal.pgen.1006170>

Tissue-specific expression and regulation of sexually dimorphic genes in mice
PMID: 16825664

We agree with the reviewer, and extended the paragraph of transcriptional sexual dimorphism in the introduction (P.4) with the suggested publications (as well as few other recent publications).

8. In this study, they re-analyze existing ATAC-seq data (IMMGEM) and point out that there are some differences in open chromatin regions between males and females – Firre, GM35612, and Pls3 (With respect to Fig.5C,D). The authors should put this finding into more of a biological context. For example (1) previous reports find that Firre escapes X inactivation and there are regulatory differences in the inactive and active X. This was determined by allele specific chip and RNA seq.

Function and evolution of local repeats in the Firre locus

PMID: 27009974

Topological organization of multichromosomal regions by the long intergenic noncoding RNA Firre

PMID: 24463464

Regulation of the ESC transcriptome by nuclear long noncoding RNAs

PMID: 26048247

The lncRNA Firre anchors the inactive X chromosome to the nucleolus by binding CTCF and maintains H3K27me3 methylation

PMID: 25887447

We added those publications to the discussion section P.17. We also updated the reference to the now published ImmGen ATAC-seq paper.

MINOR POINTS:

1. What do the 5 different modules represent? It is not clearly stated. “The highly complicated interferon response of the same cell types as here was recently parse into five modules, that are different in terms of the cells in which they are active, their regulation, their tonic interferon response, and more.”

We only use the modules to put the interferon response here in a wider context of the general interferon response. We have now edited the description of the modules creation (P.9) to add more details from the original study defining the modules (Mostafvi *et al*), namely:

"Mostafavi et al. built a regulatory network between 92 predicted regulators and 102 interferon stimulated genes, with 2,691 links connecting regulators to genes based mostly on co-expression in human and mouse response to interferon in multiple datasets. The 102 interferon stimulated genes in the regulatory network were then parsed into five distinct modules based on similarity of their regulators – C1-C5. Thus, the modules are different in terms of regulation by definition, but

turned out to be informative in that they are also distinct in the cell types in which they are active, their tonic interferon response, and more. For example, STAT1/2 and IRF9 were the main predicted regulators C3 and C4, and accordingly those modules were enriched for core ISRE motif. Module C3 contained mostly antiviral effectors and key positive and negative regulators. C1 and C2 were enriched for RNA processing, C4 for metabolic regulation, and C5 for inflammation mediators or regulators."

2. *Missing Citation: "An example of this potential effect is the X-linked gene DEAD-box helicase 3 (Ddx3x) and its Y-linked homolog (Ddx3y). Ddx3x is crucial for interferon (IFN) production in response to pathogens, and high levels of Ddx3x can boost the female IFN inducers response."*

We rephrased the sentence to clarify which information comes from which source, and added a recent publication showing sexual dimorphism in the response to infection in DDX3Y hematopoietic KO (PMID: 30475900)

" Another example of this potential effect is the X-linked gene DEAD-box helicase 3 (*Ddx3x*) and its Y-linked homolog (*Ddx3y*). *Ddx3x* is crucial for interferon (IFN) production in response to pathogens¹⁷ and in high levels can boost the female IFN inducers response. Indeed, mice lacking *Ddx3x* in hematopoietic cells have higher susceptibility to *Listeria monocytogenes* and reduced numbers of lymphocytes, not compensated by *Ddx3y*¹⁸"

Reviewer #2 (Immune ageing, systems immunology)(Remarks to the Author):

The manuscript addresses the highly relevant question of sexual dimorphism in immune responses and autoimmune. The approach taken to extend the transcriptome data base of the Immunological Genome Project to include female mice and extend this study to include ATAC-seq data and transcriptional data after in vivo challenge is timely. Data generation followed standard operating procedure developed by ImmunGen and quality control and analytical tools appear to be overall appropriate. The most striking finding is that gender differences were small, of borderline significance and not consistent across tissue, even not related myeloid tissues.

1. *Given the small differences, a discussion on the power of the study would be desirable.*

We agree – given our small sample size, especially for the cell type specific effect, we can only identify relatively large effect sizes. In the initial submission we were satisfied with that, as the known sex related genes (XIST on chr x, Eif2S3Y on chromosome Y) were clearly identified in both pan-immune analysis and cell type

specific analysis. Additional differentially expressed genes were identified in the pan immune analysis, and for macrophages. As macrophages had the same test-parameters as the other cell types, the fact that for the other cell types no SDEGs were identified is not solely due to lack of power. As also seen in the human analysis of the much larger ImmVar data, the magnitude of differences between sexes is apparently small, and requires larger sample size.

Following the reviewer comment, we learned that a formal power analysis for a multiple test setup of thousands of tests with FDR is non trivial. The power to detect a specific effect in a given test depends on the results of all tests. In our case, it means that a gene with a given t-statistic in the per-cell type setting may be found significant at one cell type and not significant in another with the same FDR threshold.

We recruited a statistician (Or Zuk) as an additional co-author to address this issue. After a literature search, we realized that power analysis for a multiple test setup of thousands of tests with FDR was only addressed in a few publications with no consensus for method of choice for power analysis. The main two papers are:

1. Efron, Bradley. "Size, power and false discovery rates." *The Annals of Statistics* 35, no. 4 (2007): 1351-1377.
2. Bi, Ran, and Peng Liu. "Sample size calculation while controlling false discovery rate for differential expression analysis with RNA-sequencing experiments." *BMC bioinformatics* 17, no. 1 (2016): 146.

Thus, an alternative for the traditional power analysis is modeling the test results distribution as a combination of the null distribution and the signal distribution. The signal distribution parameters and the fraction of tests on signal need to be estimated, as is done by Efron et al. Then, at any p-value threshold, we can use the model to calculate the FDR and the power. We added the power analysis described below as a supplementary note to the paper (Supplementary Note 1)

2. Given that gender signatures are tissue specific, the potential limitation that cells were not stratified for differentiation status should be discussed. For example, there are substantial differences between naïve and memory T cells in transcriptomes and chromatin accessibility maps.

We agree with the reviewer, and added a discussion of this point in P.18-19. However, as at the level of all T cells there is practically no transcriptional difference between male and female CD4 or CD8 T cells, going into a finer resolution of more coherent sub types of T cells (or other lineages) is unlikely to lead to more differences. Digging in for more coherent macrophages sub types is certainly a future direction, and most likely even going to single cell analysis of male and female macrophages.

3. The study included three different age groups, but age is only mentioned in passing

in the result section. If age is important variable, could the inclusion of three different age groups reduced the powered?

This point was also raised by the first reviewer. We acknowledge this point should be clarified, and clarified accordingly in the results section 'Cell-type-specific sex signature' (p.8). In both pan immune and cell type specific tests, we paired the samples by age. We added a supplementary figure of the SDEGs heatmaps with colorbars indicating age and dataset to address this point (Suppl. Figure 2A).

As Figure R1B above shows, the standard deviation of the 41 SDEGs in the three samples that are of the same age is mostly similar to that in the three samples from different ages for males. For females (Figure R1A above), the old female is a bit of an outlier (as can be seen in paper Figure 3b and Suppl. Figure 2A) – not enough to be removed, but enough to drive the standard deviation of 5 SDEGs to be higher. We also show above, two figures (Figure R1C-D) with the patterns of all 41 SDEGs (separately for male and female SDEGs). Both show that the 41 SDEGs are differentially expressed between male and female, independent of age.

However, as the reviewers predicted, some genes seem to be more different in specific ages. Unfortunately, our design cannot robustly identify such effects.

4. What was the rationale for using different FDRs for different comparisons?

Admittedly, in retrospect, several arbitrary FDR values were used with no justification. We have narrowed down the number of different threshold, justified below:

One FDR threshold for all male-female differential gene expression analyses in mice (FDR <0.2). Pretty liberal, as very few SDEGs identified.

Second FDR Threshold (FDR < 0.05) was used for the more robust effects: Differential gene expression after IFN stimulation, Differential gene expression between males and females in humans, differential OCRs and functional enrichment analysis (on tissue MF ANOVA results).

The pan-immune differential pathways analysis was done by the GSEA software which uses 25% FDR as the default cutoff.

Correction were made throughout the text when the adjustment of FDR threshold changed the results. No qualitative results were changed. Specific changes:

1. The results of functional enrichment analysis of the sex*tissue interaction in macrophages. Some annotations were removed (correction in P.13 & Supp. table 9).
2. The sentence "Finding 14 SDEGs is more than expected by chance (permutation p value = 0.035)." was removed, as the permutation p-value with FDR of 0.2 is now 0.06.

5. *Although of interest, the comparison to SLE transcriptome analysis remains superficial. It is unclear whether the two chosen comparison datasets are representative. There is no discussion on the heterogeneity of cell types in PBMC that account for the SLE signature in the chosen datasets and the SLE score is not well explained.*

We accept the reviewer comment and add clarification about the studies and the SLE score (Results section P.9). Of note, SLE score was not defined by us but by Mostafavi et al. (Cell, 2016).

We also add a discussion about the limitation of comparing PBMC and sorted MFs to the discussion section (P.16)

6. *The findings on Firre are of interest, but preliminary and should not be overinterpreted. It is certainly not justified to make Firre a focus point in the abstract.*

We accept and remove Firre mention from the abstract.

7. *The data appear to be consistent with the interpretation that female cells are less able to maintain quiescence and macrophages are the more responsive cell type. Are the observed signatures truly an IFN response signature or do they just reflect macrophage activation?*

We searched for macrophage activation signature, and compared against our 41 SDEGs. We compared our 41 SDEGs to the M1 (inflammation) and M2 (tissue repair) transcriptional modules published by Joachim Shultze's group (Immunity, 2014). There was no overlap between the lists, besides one gene, Serpinb2, which belong to the M2 module and identified in our study as male specific gene in macrophages.

8. *The authors should discuss that the human studies but not the mouse studies show a signature for T cells. The manuscript by Howard Chung's group in Cell Systems that ATAC-seq signatures in PBMCs are driven by gender should be discussed.*

Indeed, Howard Chung's group have identified Firre as more accessible in females T4 cells, which is in agreement with our findings in mice, and in MFs. This point was added to the discussion in page 17.

Minor comments

Incomplete sentence in the introduction (Landscape of X chromosome inactivation across human tissues).

Apologies. Half sentence removed.

REVIEWERS' COMMENTS:

Reviewer #1 (Remarks to the Author):

The authors have addressed all my suggestions above and beyond sufficiency. I thank the authors and congratulate them on a fantastic resource and approach.

Reviewer #2 (Remarks to the Author):

The authors have appropriately addressed my comments.

Reviewer #3 (Remarks to the Author):

My comments are about the power analysis raised in the first point in reviewer2's report.

I felt that the author's response is adequate and reasonable. What the authors reported in the manuscript is an exploratory study, different from the traditional hypothesis-driven type of research work. In this type of study, opportunistic discoveries were made given the constraint of the study set up. Often the ones with large effect size (low hanging fruits) were found. In my opinion, power and sample size calculation is designed for studies with a hypothesis fixed a priori. Therefore, i felt that a traditional power analysis is not very helpful for the current study, although it still can be done. What the authors have done in response to the reviewer's comment is reasonable. The approach they cited from Dr. Efron is well accepted in the statistics community. Therefore, I am satisfied with the authors' response and do not have concerns on the method they used for power analysis.